# Planning in entropy-regularized
# Markov decision processes and games

**Jean-Bastien Grill**[*]
DeepMind Paris
jbgrill@google.com

**Omar D. Domingues**[*]
SequeL team, Inria Lille
omar.darwiche-domingues@inria.fr

**Pierre Ménard**
SequeL team, Inria Lille
pierre.menard@inria.fr

**Rémi Munos**
DeepMind Paris
munos@google.com

**Michal Valko**
DeepMind Paris
valkom@deepmind.com

## Abstract

We propose `SmoothCruiser`, a new planning algorithm for estimating the value function in entropy-regularized Markov decision processes and two-player games, given a generative model of the environment. `SmoothCruiser` makes use of the smoothness of the Bellman operator promoted by the regularization to achieve *problem-independent sample complexity* of order $\widetilde{\mathcal{O}}(1/\varepsilon^4)$ for a desired accuracy $\varepsilon$, whereas for non-regularized settings there are no known algorithms with guaranteed polynomial sample complexity in the worst case.

## 1 Introduction

Planning with a generative model is *thinking before acting*. An agent thinks using a world model that it has built from prior experience [Sutton, 1991, Sutton and Barto, 2018]. In the present paper, we study planning in two types of environments, *Markov decision processes* (MDPs) and *two-player turn-based zero-sum games*. In both settings, agents interact with an environment by taking actions and receiving rewards. Each action changes the state of the environment and the agent aims to choose actions to maximize the sum of rewards. We assume that we are given a generative model of the environment, that takes as input a state and an action and returns a reward and a next state as output. Such generative models, called *oracles*, are typically built from known data and involve simulations, for example, a physics simulation. In many cases, simulations are costly. For example, simulations may require the computation of approximate solutions of differential equations or the discretization of continuous state spaces. Therefore, a smart algorithm *makes only a small the number of oracles calls* required to estimate the value of a state. The total number of oracle calls made by an algorithm is referred to as *sample complexity*.

The value of a state $s$, denoted by $V(s)$, is the maximum of the sum of discounted rewards that can be obtained from that state. We want an algorithm that returns an estimate of precision $\varepsilon$ of the $V(s)$ for any fixed $s$ and has a low sample complexity, which should naturally be a function of $\varepsilon$. An agent can then use this algorithm to predict the value of the possible actions at any given state and choose the best one. The main advantage in estimating the value of a *single* given state $s$ at a time instead of the complete value function[2] $s \mapsto V(s)$ is that we can have algorithms whose sample complexity does not depend on the size of the state space, which is important when our state space is very large or continuous. On the other hand, the disadvantage is that the algorithm must be run each time a new state is encountered.

---

[*]equal contribution

[2]as done by approximate dynamic programming

Our main contribution is an algorithm that *estimates the value function in a given state* in planning problems that satisfy specific smoothness conditions, which is the case when the rewards are regularized by adding an entropy term. We exploit this smoothness property to obtain a polynomial sample complexity of order $\widetilde{\mathcal{O}}\big(1/\varepsilon^4\big)$ that is *problem independent*.

**Related work**  Kearns et al. [1999] came up with a sparse sampling algorithm (SSA) for planning in MDPs with finite actions and arbitrary state spaces. SSA estimates the value of a state $s$ by building a sparse look-ahead tree starting from $s$. However, SSA achieves a sample complexity of $\mathcal{O}\big((1/\varepsilon)^{\log(1/\varepsilon)}\big)$, which is non-polynomial in $1/\varepsilon$. SSA is slow since its search is *uniform*, i.e., it does not select actions adaptively. Walsh et al. [2010] gave an improved version of SSA with adaptive action selection, but its sample complexity is still non-polynomial. The UCT algorithm [Kocsis and Szepesvári, 2006], used for planning in MDPs and games, selects actions based on optimistic estimates of their values and has good empirical performance in several applications. However, the sample complexity of UCT can be worse than exponential in $1/\varepsilon$ for some environments, which is mainly due to exploration issues [Coquelin and Munos, 2007]. Algorithms with sample complexities of order $\mathcal{O}\big(1/\varepsilon^d\big)$, where $d$ is a problem-dependent quantity, have been proposed for deterministic dynamics [Hren and Munos, 2008], and in an open-loop[3] setting [Bubeck and Munos, 2010, Leurent and Maillard, 2019, Bartlett et al., 2019], for bounded number of next states and a full MDP model is known [Buşoniu and Munos, 2012], for bounded number of next states in a *finite-horizon* setting [Feldman and Domshlak, 2014], for bounded number of next states [Szörényi et al., 2014], and for general MDPs [Grill et al., 2016]. In general, when the state space is infinite and the transitions are stochastic, the problem-dependent quantity $d$ can make the sample complexity guarantees exponential. For a related setting, when rewards are only obtained in the leaves of a fixed tree, Kaufmann and Koolen [2017] and Huang et al. [2017] present algorithms to identify the optimal action in a game based on best-arm identification tools.

Entropy regularization in MDPs and reinforcement learning have been employed in several commonly used algorithms. In the context of policy gradient algorithms, common examples are the TRPO algorithm [Schulman et al., 2015] which uses the Kullback-Leibler divergence between the current and the updated policy to constrain the gradient step sizes, the A3C algorithm [Mnih et al., 2016] that penalizes policies with low entropy to improve exploration, and the work of Neu et al. [2017] presenting a theoretical framework for entropy regularization using the joint state-action distribution. Formulations with entropy-augmented rewards, which is the case in our work, have been used to learn multi-modal policies to improve exploration and robustness [Haarnoja et al., 2017, 2018] and can also be related to policy gradient methods [Schulman et al., 2017]. Furthermore, Geist et al. [2019] propose a theory of regularized MDPs which includes entropy as a special case. Summing up, reinforcement learning knows *how* to employ entropy regularization. In this work, we tasked ourselves to give insights on *why*.

## 2   Setting and motivation

Both MDPs and two-player games can be formalized as a tuple $(\mathcal{S}, \mathcal{A}, P, R, \gamma)$, where $\mathcal{S}$ is the set of states, $\mathcal{A}$ is the set of actions, $P \triangleq \{P(\cdot|s,a)\}_{s,a \in \mathcal{S} \times \mathcal{A}}$ is a set of probability distributions over $\mathcal{S}$, $R : \mathcal{S} \times \mathcal{A} \to [0,1]$ is a (possibly random) reward function and $\gamma \in [0,1[$ is the known discount factor. In the MDP case, at each round $t$, an agent is at state $s$, chooses action $a$ and observes a reward $R(s,a)$ and a transition to a next state $z \sim P(\cdot|s,a)$. In the case of turn-based two-player games, there are two agents and, at each round $t$, an agent chooses an action, observes a reward and a transition; at round $t+1$ it's the other player's turn. This is equivalent to an MDP with an augmented state space $\mathcal{S}^+ \triangleq \mathcal{S} \times \{1,2\}$ and transition probabilities such that $P((z,j)|(s,i),a) = 0$ if $i = j$. We assume that the action space $\mathcal{A}$ is finite with cardinality $K$ and the state space $\mathcal{S}$ has arbitrary (possibly infinite) cardinality.

Our objective is to find an algorithm that outputs a good estimate of the value $V(s)$ for any given state $s$ as quickly as possible. An agent can then use this algorithm to choose the best action in an MDP or a game. More precisely, for a state $s \in \mathcal{S}$ and given $\varepsilon > 0$ and $\delta > 0$, our goal is to compute an estimate $\widehat{V}(s)$ of $V(s)$ such that $\mathbb{P}\Big[\big|\widehat{V}(s) - V(s)\big| > \varepsilon\Big] \leq \delta$ with small number of oracle calls

required to compute this estimate. In our setting, we consider the case of *entropy-regularized* MDPs and games, where the objective is augmented with an entropy term.

## 2.1 Value functions

**Markov decision process**   The policy $\pi$ of an agent is a function from $\mathcal{S}$ to $\mathcal{P}(\mathcal{A})$, the set of probability distributions over $\mathcal{A}$. We denote by $\pi(a|s)$ the probability of the agent choosing action $a$ at state $s$. In MDPs, the value function at a state $s$, $V(s)$, is defined as the supremum over all possible policies of the expected sum of discounted rewards obtained starting from $s$, which satisfies the Bellman equations [Puterman, 1994],

$$\forall s \in \mathcal{S}, \ V(s) = \max_{\pi(\cdot|s)\in\mathcal{P}(\mathcal{A})} \mathbb{E}[R(s,a) + \gamma V(z)], \ a \sim \pi(\cdot|s), \ z \sim P(\cdot|s,a). \tag{1}$$

**Two-player turn-based zero-sum games**   In this case, there are two agents (1 and 2), each one with its own policy and different goals. If the policy of Agent 2 is fixed, Agent 1 aims to find a policy that *maximizes* the sum of discounted rewards. Conversely, if the policy of Agent 1 is fixed, Agent 2 aims to find a policy that *minimizes* this sum. Optimal strategies for both agents can be shown to exist and for any $(s,i) \in \mathcal{S}^+ \triangleq \mathcal{S} \times \{1,2\}$, the value function is defined as [Hansen et al., 2013]

$$V(s,i) \triangleq \begin{cases} \max_{\pi(\cdot|s)\in\mathcal{P}(\mathcal{A})} \mathbb{E}[R((s,i),a) + \gamma V(z,j)], & \text{if } i = 1, \\ \min_{\pi(\cdot|s)\in\mathcal{P}(\mathcal{A})} \mathbb{E}[R((s,i),a) + \gamma V(z,j)], & \text{if } i = 2, \end{cases} \tag{2}$$

with $a \sim \pi(\cdot|s)$ and $(z,j) \sim P(\cdot|(s,i),a)$. In this case, the function $s \mapsto V(s,i)$ is the optimal value function for Agent $i$ when the other agent follows its optimal strategy.

**Entropy-regularized value functions**   Consider a regularization factor $\lambda > 0$. In the case of MDPs, when rewards are augmented by an entropy term, the value function at state $s$ is given by [Haarnoja et al., 2017, Dai et al., 2018, Geist et al., 2019]

$$V(s) \triangleq \max_{\pi(\cdot|s)\in\mathcal{P}(\mathcal{A})} \left\{ \mathbb{E}[R(s,a) + \gamma V(z)] + \lambda \mathcal{H}(\pi(\cdot|s)) \right\}, \ a \sim \pi(\cdot|s), \ z \sim P(\cdot|s,a)$$

$$= \lambda \log \sum_{a\in\mathcal{A}} \exp\left(\tfrac{1}{\lambda}\mathbb{E}[R(s,a) + \gamma V(z)]\right), \ z \sim P(\cdot|s,a), \tag{3}$$

where $\mathcal{H}(\pi(\cdot|s))$ is the entropy of the probability distribution $\pi(\cdot|s) \in \mathcal{P}(\mathcal{A})$.

The function $\text{LogSumExp}_\lambda : \mathbb{R}^K \to \mathbb{R}$, defined as $\text{LogSumExp}_\lambda(x) \triangleq \lambda \log \sum_{i=1}^K \exp(x_i/\lambda)$, is a smooth approximation of the $\max$ function, since $\|\max - \text{LogSumExp}_\lambda\|_\infty \leq \lambda \log K$. Similarly, the function $-\text{LogSumExp}_{-\lambda}$ is a smooth approximation of the $\min$ function. This allows us to define the regularized version of the value function for turn-based two player games, in which both players have regularized rewards, by replacing the $\max$ and the $\min$ in Equation 2 by their smooth approximations.

For any state $s$, let $F_s \triangleq \text{LogSumExp}_\lambda$ or $F_s \triangleq -\text{LogSumExp}_{-\lambda}$ depending on $s$. Both for MDPs and games, we can write the entropy-regularized value functions as

$$V(s) = F_s(Q_s), \ \text{with} \ Q_s(a) \triangleq \mathbb{E}[R(s,a) + \gamma V(z)], \ z \sim P(\cdot|s,a), \tag{4}$$

where $Q_s \triangleq (Q_s(a))_{a\in\mathcal{A}}$, the $Q$ function at state $s$, is a vector in $\mathbb{R}^K$ representing the value of each action. The function $F_s$ is the *Bellman operator* at state $s$, which becomes smooth due to the entropy regularization.

**Useful properties**   Our algorithm exploits the smoothness property of $F_s$ defined above. In particular, these functions are $L$-smooth, that is, for any $Q, Q' \in \mathbb{R}^K$, we have

$$|F_s(Q) - F_s(Q') - (Q - Q')^\top \nabla F_s(Q')| \leq L\|Q - Q'\|_2^2, \ \text{with} \ L = 1/\lambda \cdot \tag{5}$$

Furthermore, the functions $F_s$ have two important properties: $\nabla F_s(Q)$[4] $\succeq 0$ and $\|\nabla F_s(Q)\|_1 = 1$ for all $Q \in \mathbb{R}^K$. This implies that the gradient $\nabla F_s(Q)$ defines a probability distribution.[5]

**Assumptions**  We assume that $\mathcal{S}$, $\mathcal{A}$, $\lambda$, and $\gamma$ are given to the learner. Moreover, we assume that we can access a generative model, the *oracle*, from which we can get reward and transition samples from arbitrary state-action pairs. Formally, when called with parameter $(s, a) \in \mathcal{S} \times \mathcal{A}$, the oracle outputs a new random variable $(R, Z)$ independent from any other outputs received from the generative model so far such that $Z \sim P(\cdot|s, a)$ and $R$ has same distribution as $R(s, a)$. We denote a call to the oracle as $R, Z \leftarrow \texttt{oracle}(s, a)$.

## 2.2   Using regularization for the polynomial sample complexity

To pave the road for $\texttt{SmoothCruiser}$, we consider two extreme cases, based on the strength of the regularization:

1. **Strong regularization**  In this case, $\lambda \to \infty$ and $L = 0$, that is, $F_s$ is linear for all $s$: $F_s(x) = w_s^\intercal x$, with $\|w_s\|_1 = 1$, $w_s \in \mathbb{R}^k$ and $w_s \succeq 0$,

2. **No regularization**  In this case, $\lambda = 0$ and $L \to \infty$, that is, $F_s$ cannot be well approximated by a linear function.[6]

In the strongly regularized case, we can approximate the value $V(s)$ with $\widetilde{\mathcal{O}}(1/\varepsilon^2)$ oracle calls. This is due to the linearity of $F_s$, since the value function can be written as $V(s) = \mathbb{E}[\sum_{t=0}^\infty \gamma^t R(S_t, A_t) \mid S_0 = s]$ where $A_t$ is distributed according to the probability vector $w_{S_t}$. As a result, $V(s)$ can be estimated by Monte-Carlo sampling of trajectories.

With no regularization, we can apply a simple adaptation of the sparse sampling algorithm of Kearns et al. [1999] that we briefly describe. Assume that we have an subroutine that provides an approximation of the value function with precision $\varepsilon/\sqrt{\gamma}$, denoted by $\widehat{V}_{\varepsilon/\sqrt{\gamma}}(s)$, for any $s$. We can call this subroutine several times as well as the oracle to get improved estimate $\widehat{V}$ defined as

$$\widehat{V}(s) = F_s\left(\widehat{Q}_s\right) \quad \text{with} \quad \widehat{Q}_s(a) \leftarrow \frac{1}{N} \sum_{i=1}^N \Big[ r_i(s, a) + \gamma \widehat{V}_{\varepsilon/\sqrt{\gamma}}(z_i) \Big],$$

where $r_i(s, a)$ and $z_i$ are rewards and next states sampled by calling the oracle with parameters $(s, a)$. By Hoeffding's inequality, we can choose $N = \mathcal{O}(1/\varepsilon^2)$ such that $\widehat{V}(s)$ is an approximation of $V(s)$ with precision $\varepsilon$ with high probability. By applying this idea recursively, we start with $\widehat{V} = 0$, which is an approximation of the value function with precision $1/(1 - \gamma)$, and progressively improve the estimates towards a desired precision $\varepsilon$, which can be reached at a recursion depth of $H = \mathcal{O}(\log(1/\varepsilon))$. Following the same reasoning as Kearns et al. [1999], this approach has a sample complexity of $\mathcal{O}\big((1/\varepsilon)^{\log(1/\varepsilon)}\big)$: to estimate the value at a given recursion depth, we make $\mathcal{O}(1/\varepsilon^2)$ recursive calls and stop once we reach the maximum depth, resulting in a sample complexity of

$$\underbrace{\frac{1}{\varepsilon^2} \times \cdots \times \frac{1}{\varepsilon^2}}_{\mathcal{O}(\log(1/\varepsilon)) \text{ times}} = \left(\frac{1}{\varepsilon}\right)^{\mathcal{O}\big(\log\big(\frac{1}{\varepsilon}\big)\big)}.$$

In the next section, we provide $\texttt{SmoothCruiser}$ (Algorithm 1), that uses the assumption that the functions $F_s$ are $L$-smooth with $0 < L < \infty$ to interpolate between the two cases above and obtain a sample complexity of $\widetilde{\mathcal{O}}(1/\varepsilon^4)$.

## 3   SmoothCruiser

We now describe our planning algorithm. Its building blocks are two procedures, $\texttt{sampleV}$ (Algorithm 2) and $\texttt{estimateQ}$ (Algorithm 3) that recursively call each other. The procedure $\texttt{sampleV}$ returns a noisy estimate of $V(s)$ with a bias bounded by $\varepsilon$. The procedure $\texttt{estimateQ}$ averages the outputs of several calls to $\texttt{sampleV}$ to obtain an estimate $\widehat{Q}_s$ that is an approximation of $Q_s$ with precision $\varepsilon$ with high probability. Finally, $\texttt{SmoothCruiser}$ calls $\texttt{estimateQ}(s, \varepsilon)$ to obtain $\widehat{Q}_s$ and outputs $\widehat{V}(s) = F_s(\widehat{Q}_s)$. Using the assumption that $F_s$ is 1-Lipschitz, we can show that $\widehat{V}(s)$ is an approximation of $V(s)$ with precision $\varepsilon$. Figure 1 illustrates a call to $\texttt{SmoothCruiser}$.

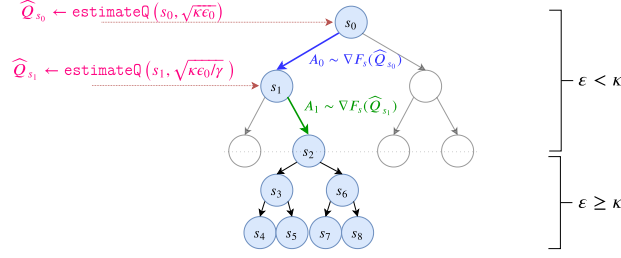

$\widehat{Q}_{s_0} \leftarrow \texttt{estimateQ}\left(s_0, \sqrt{\kappa\epsilon_0}\right)$

$\widehat{Q}_{s_1} \leftarrow \texttt{estimateQ}\left(s_1, \sqrt{\kappa\epsilon_0/\gamma}\right)$

$A_0 \sim \nabla F_s(\widehat{Q}_{s_0})$

$A_1 \sim \nabla F_s(\widehat{Q}_{s_1})$

$\varepsilon < \kappa$

$\varepsilon \geq \kappa$

Figure 1: Visualization of a call to $\texttt{SmoothCruiser}(s_0, \varepsilon_0, \delta')$.

## 3.1 Smooth sailing

The most important part of the algorithm is the procedure $\texttt{sampleV}$, that returns a low-bias estimate of the value function. Having the estimate of the value function, the procedure $\texttt{estimateQ}$ averages the outputs of $\texttt{sampleV}$ to obtain a good estimate of the $Q$ function with high probability. The main idea of $\texttt{sampleV}$ is to first compute an estimate of precision $\mathcal{O}(\sqrt{\varepsilon})$ of the value of each action $\{\widehat{Q}_s(a)\}_{a \in \mathcal{A}}$ to linearly approximate the function $F_s$ around $\widehat{Q}_s$.

---

**Algorithm 1** $\texttt{SmoothCruiser}$

**Input:** $(s, \varepsilon, \delta') \in \mathcal{S} \times \mathbb{R}_+ \times \mathbb{R}_+$
$M_\lambda \leftarrow \sup_{s \in \mathcal{S}} |F_s(0)| = \lambda \log K$
$\kappa \leftarrow (1 - \sqrt{\gamma})/(KL)$
Set $\delta'$, $\kappa$, and $M_\lambda$ as global parameters
$\widehat{Q}_s \leftarrow \texttt{estimateQ}(s, \varepsilon)$
**Output:** $F_s\left(\widehat{Q}_s\right)$

---

The local approximation of $F_s$ around $\widehat{Q}_s$ is subsequently used to estimate the value of $s$ with a better precision, of order $\mathcal{O}(\varepsilon)$, which is possible due to the smoothness of $F_s$.

---

**Algorithm 2** $\texttt{sampleV}$

1: **Input:** $(s, \varepsilon) \in \mathcal{S} \times \mathbb{R}_+$
2: **if** $\varepsilon \geq (1 + M_\lambda)/(1 - \gamma)$ **then**
3:     **Output:** 0
4: **else if** $\varepsilon \geq \kappa$ **then**
5:     $\widehat{Q}_s \leftarrow \texttt{estimateQ}(s, \varepsilon)$
6:     **Output:** $F_s\left(\widehat{Q}_s\right)$
7: **else if** $\varepsilon < \kappa$ **then**
8:     $\widehat{Q}_s \leftarrow \texttt{estimateQ}(s, \sqrt{\kappa\varepsilon})$
9:     $A \leftarrow$ action drawn from $\nabla F_s\left(\widehat{Q}_s\right)$
10:     $(R, Z) \leftarrow \texttt{oracle}(s, A)$
11:     $\widehat{V} \leftarrow \texttt{sampleV}(Z, \varepsilon/\sqrt{\gamma})$
12:     **Output:**
13:       $F_s\left(\widehat{Q}_s\right) - \widehat{Q}_s^\mathsf{T} \nabla F_s\left(\widehat{Q}_s\right) + (R + \gamma\widehat{V})$
14: **end if**

---

**Algorithm 3** $\texttt{estimateQ}$

1: **Input:** $(s, \varepsilon)$
2: $N(\varepsilon) \leftarrow \left\lceil \frac{18(1+M_\lambda)^2}{(1-\gamma)^4(1-\sqrt{\gamma})^2} \frac{\log\left(2K/\delta'\right)}{\varepsilon^2} \right\rceil$
3: **for** $a \in \mathcal{A}$ **do**
4:     $q_i \leftarrow 0$ for $i \in 1, ..., N(\varepsilon)$
5:     **for** $i \in 1, ..., N(\varepsilon)$ **do**
6:       $(R, Z) \leftarrow \texttt{oracle}(s, a)$.
7:       $\widehat{V} \leftarrow \texttt{sampleV}\left(Z, \varepsilon/\sqrt{\gamma}\right)$
8:       $q_i \leftarrow R + \gamma\widehat{V}$
9:     **end for**
10:     $\widehat{Q}_s(a) \leftarrow \textbf{mean}(q_1, \ldots, q_N)$
11:     // clip $\widehat{Q}_s(a)$ to $[0, (1 + M_\lambda)/(1 - \gamma)]$
12:     $\widehat{Q}_s(a) \leftarrow \max(0, \widehat{Q}_s(a))$
13:     $\widehat{Q}_s(a) \leftarrow \min((1+M_\lambda)/(1-\gamma), \widehat{Q}_s(a))$
14: **end for**
15: **Output:** $\widehat{Q}_s$

---

For a target accuracy $\varepsilon$ at state $s$, $\texttt{sampleV}$ distinguishes three cases, based on a reference threshold $\kappa \triangleq (1 - \sqrt{\gamma})/(KL)$, which is the maximum value of $\varepsilon$ for which we can compute a good estimate of the value function using linear approximations of $F_s$.

- **First,** if $\varepsilon \geq (1 + \lambda \log K)/(1 - \gamma)$, then 0 is a valid output, since $V(s)$ is bounded by $(1 + \lambda \log K)/(1 - \gamma)$. This case furthermore ensures that our algorithm terminates, since the recursive calls are made with increasing values of $\varepsilon$.

- **Second,** if $\kappa \leq \varepsilon \leq (1 + \lambda \log K)/(1 - \gamma)$, we run $F_s(\texttt{estimateQ}(s, \varepsilon))$ in which for each action, both the oracle and $\texttt{sampleV}$ are called $\mathcal{O}(1/\varepsilon^2)$ times in order to return $\widehat{V}(s)$ which is with high probability an $\varepsilon$-approximation of $V(s)$.

- **Finally,** if $\varepsilon < \kappa$, we take advantage of the smoothness of $F_s$ to compute an $\varepsilon$-approximation of $V(s)$ in a more efficient way than calling the oracle and `sampleV` $\mathcal{O}(1/\varepsilon^2)$ times. We achieve it by calling `estimateQ` with a precision $\sqrt{\kappa\varepsilon}$ instead of $\varepsilon$, which requires $\mathcal{O}(1/\varepsilon)$ calls instead.

## 3.2 Smoothness guarantee an improved sample complexity

In this part, we describe the key ideas that allows us to exploit the smoothness of the Bellman operator to obtain a better sample complexity. Notice that when $\varepsilon < \kappa$, the procedure `estimateQ` is called to obtain an estimate $\widehat{Q}_s$ such that

$$\|\widehat{Q}_s - Q_s\|_2 = \mathcal{O}\left(\sqrt{\varepsilon/L}\right).$$

The procedure `sampleV` then continues with computing a linear approximation of $F_s(Q_s)$ around $\widehat{Q}_s$. Using the $L$-smoothness of $F_s$, we guarantee the $\varepsilon$-approximation,

$$|F_s(Q_s) - \left\{ F_s(\widehat{Q}_s) + (Q_s - \widehat{Q}_s)^{\mathsf{T}} \nabla F_s(\widehat{Q}_s) \right\}| \leq L\|\widehat{Q}_s - Q_s\|_2^2 = \mathcal{O}(\varepsilon).$$

We wish to output this linear approximation, but we need to handle the fact that the vector $Q_s$ (the true $Q$-function at $s$) is unknown. Notice that the vector $\nabla F_s(\widehat{Q}_s)$ represents a probability distribution. The term $Q_s^{\mathsf{T}} \nabla F_s(\widehat{Q}_s)$ in the linear approximation of $F_s(Q_s)$ above can be expressed as

$$Q_s^{\mathsf{T}} \nabla F_s(\widehat{Q}_s) = \mathbb{E}\left[ Q_s(A) \Big| \widehat{Q}_s \right], \text{ with } A \sim \nabla F_s(\widehat{Q}_s).$$

Therefore, we can build a low-bias estimate of $Q_s^{\mathsf{T}} \nabla F_s(\widehat{Q}_s)$ from estimating only $Q_s(A)$:

- sample action $A \sim \nabla F_s(\widehat{Q}_s)$
- call the generative model to sample a reward and a next state $R_{s,A}, Z_{s,A} \leftarrow \mathtt{oracle}(s, A)$
- obtain an $\mathcal{O}(\varepsilon)$-approximation of $Q_s(A)$: $\widetilde{Q}(A) = R_{s,A} + \gamma \mathtt{sampleV}\left(Z_{s,A}, \varepsilon/\sqrt{\gamma}\right)$
- output $\widehat{V}(s) = F_s(\widehat{Q}_s) - \widehat{Q}_s^{\mathsf{T}} \nabla F_s(\widehat{Q}_s) + \widetilde{Q}(A)$

We show that $\widehat{V}(s)$ is an $\varepsilon$-approximation of the true value function $V(s)$. The benefit of such approach is that we can call `estimateQ` with a precision $\mathcal{O}(\sqrt{\varepsilon})$ instead of $\mathcal{O}(\varepsilon)$, which thanks to the smoothness of $F_s$, reduces the sample complexity. In particular, one call to `sampleV`$(s, \varepsilon)$ will make $\mathcal{O}(1/\varepsilon)$ recursive calls to `sampleV`$(s, \mathcal{O}(\sqrt{\varepsilon}))$, and the total number of calls to `sampleV` behaves as

$$\frac{1}{\varepsilon} \times \frac{1}{\varepsilon^{1/2}} \times \frac{1}{\varepsilon^{1/4}} \times \cdots \leq \frac{1}{\varepsilon^2}.$$

Therefore, the number of `sampleV` calls made by `SmoothCruiser` is of order $\mathcal{O}(1/\varepsilon^2)$, which implies that the total sample complexity is of $\mathcal{O}(1/\varepsilon^4)$.

## 3.3 Comparison to Monte-Carlo tree search

Several planning algorithms are based on Monte-Carlo tree search (MCTS, Coulom, 2007, Kocsis and Szepesvári, 2006). Algorithm 4 gives a template for MCTS, which uses the procedure `search` that calls `selectAction` and `evaluateLeaf`. Algorithm 5, `search`, returns an estimate of the value function; `selectAction` selects the action to be executed

---

**Algorithm 4** `genericMCTS`

---
**Input:** state $s$
**repeat** `search`$(s, 0)$
**until** timeout
**Output:** estimate of best action or value.

---

(also called *tree policy*); and `evaluateLeaf` returns an estimate of the value of a leaf. We now provide the analogies that make it possible to see `SmoothCruiser` as an MCTS algorithm:

- `sampleV` corresponds to the function `search`

- `selectAction` is implemented by calling `estimateQ` to compute $\widehat{Q}_s$, followed by sampling an action with probability proportional to $\nabla F_s(\widehat{Q}_s)$

- `evaluateLeaf` is implemented using the sparse sampling strategy of Kearns et al. [1999], if we see leaves as the nodes reached when $\varepsilon \geq \kappa$

## 4   Theoretical guarantees

In Theorem 1 we bound the sample complexity. Note that `SmoothCruiser` is non-adaptive, hence its sample complexity is *deterministic* and *problem independent*. Indeed, since our algorithm is agnostic to the output of the oracle, it performs the same number of oracle calls for any given $\varepsilon$ and $\delta'$, regardless of the *random* outcome of these calls.

---

**Algorithm 5** `search`

---

**Input:** state $s$, depth $d$
**if** $d > d_{\max}$ **then**
    **Output:** `evaluateLeaf`$(s)$
**end if**
$a \leftarrow$ `selectAction`$(s, d)$
$R, Z \leftarrow$ `oracle`$(s, a)$
**Output:** $R + \gamma$`search`$(Z, d+1)$

---

**Theorem 1.** *Let $n(\varepsilon, \delta')$ be the number of oracle calls before* `SmoothCruiser` *terminates. For any state $s \in \mathcal{S}$ and $\varepsilon, \delta' > 0$,*

$$n(\varepsilon, \delta') \leq \frac{c_1}{\varepsilon^4} \log\left(\frac{c_2}{\delta'}\right)\left[c_3 \log\left(\frac{c_4}{\varepsilon}\right)\right]^{\log_2\left(c_5\left(\log\left(\frac{c_2}{\delta'}\right)\right)\right)} = \widetilde{\mathcal{O}}\left(\frac{1}{\varepsilon^4}\right),$$

*where $c_1, c_2, c_3, c_4$, and $c_5$ are constants that depend only on $K$, $L$, and $\gamma$.*

The proof of Theorem 1 with the exact constants is in the appendix. In Theorem 2, we provide our consistency result, stating that the output of `SmoothCruiser` applied to a state $s \in \mathcal{S}$ is a good approximation of $V(s)$ with high probability.

**Theorem 2.** *For any $s \in \mathcal{S}$, $\varepsilon > 0$, and $\delta > 0$, there exists a $\delta'$ that depends on $\varepsilon$ and $\delta$ such that the output $\widehat{V}(s)$ of* `SmoothCruiser`$(s, \varepsilon, \delta')$ *satisfies*

$$\mathbb{P}\left[\left|\widehat{V}(s) - V(s)\right| > \varepsilon\right] \leq \delta.$$

*and such that $n(\varepsilon, \delta') = \mathcal{O}\left(1/\varepsilon^{4+c}\right)$ for any $c > 0$.*

More precisely, in the proof of Theorem 2, we establish that

$$\mathbb{P}\left[\left|\widehat{V}(s) - V(s)\right| > \varepsilon\right] \leq \delta' n(\varepsilon, \delta').$$

Therefore, for any parameter $\delta'$ satisfying $\delta' n(\varepsilon, \delta') \leq \delta$, `SmoothCruiser` with parameters $\varepsilon$ and $\delta'$ provides an approximation of $V(s)$ which is $(\varepsilon, \delta)$ correct.

**Impact of regularization constant**   For a regularization constant $\lambda$, the smoothness constant is $L = 1/\lambda$. in Theorem 1 we did not make the dependence on $L$ explicit to preserve simplicity. However, it easy to analyze the sample complexity in the two limits:

> **strong regularization** $L \to 0$ and $F_s$ is linear
>
> **no regularization** $L \to \infty$ and $F_s$ is not smooth

As $L \to 0$, the condition $\kappa \leq \varepsilon \leq (1 + \lambda \log K)/(1 - \gamma)$ will be met less and eventually the algorithm will sample $N = \mathcal{O}\left(1/\varepsilon^2\right)$ trajectories, which implies a sample complexity of order $\mathcal{O}\left(1/\varepsilon^2\right)$. On the other hand, as $L$ goes to $\infty$, the condition $\varepsilon < \kappa$ will be met less and the algorithm eventually runs a uniform sampling strategy of Kearns et al. [1999], which results in a sample complexity of order $\mathcal{O}\left((1/\varepsilon)^{\log(1/\varepsilon)}\right)$, which is non-polynomial in $1/\varepsilon$.

Let $V_\lambda(s)$ be the entropy regularized value function and $V_0(s)$ be its non-regularized version. Since $F_s$ is 1-Lipschitz and $\|\text{LogSumExp}_\lambda - \max\|_\infty \leq \lambda \log K$, we can prove that $\sup_s |V_\lambda(s) - V_0(s)| \leq \lambda \log K/(1 - \gamma)$. Thus, we can interpret $V_\lambda(s)$ as an approximate value function which we can estimate faster.

**Comparison to lower bound** For non-regularized problems, Kearns et al. [1999] prove a sample complexity lower bound of $\Omega\big((1/\varepsilon)^{1/\log(1/\gamma)}\big)$, which is polynomial in $1/\varepsilon$, but its exponent grows as $\gamma$ approaches 1. For regularized problems, Theorem 1 shows that the sample complexity is polynomial with an exponent that is *independent* of $\gamma$. Hence, when $\gamma$ is close to 1, regularization gives us a better asymptotic behavior with respect to $1/\varepsilon$ than the lower bound for the non-regularized case, although we are not estimating the same value.

## 5   Generalization of SmoothCruiser

Consider the general definition of value functions in Equation 4. Although we focused on the case where $F_s$ is the LogSumExp function, which arises as a consequence of entropy regularization, our theoretical results hold for any set of functions $\{F_s\}_{s \in \mathcal{S}}$ that for any $s$ satisfy the following conditions:

1. $F_s$ is differentiable
2. $\forall Q \in \mathbb{R}^K, 0 < \|\nabla F_s(Q)\|_1 \le 1$
3. (nonnegative gradient) $\forall Q \in \mathbb{R}^K, \nabla F_s(Q) \succeq 0$
4. ($L$-smooth) there exists $L \ge 0$ such that for any $Q, Q' \in \mathbb{R}^K$
$$|F_s(Q) - F_s(Q') - (Q - Q')^\mathsf{T}\nabla F_s(Q')| \le L\|Q - Q'\|_2^2$$

For the more general definition above, we need to make two simple modifications of the procedure `sampleV`. When $\varepsilon < \kappa$, the action $A$ in `sampleV` is sampled according to

$$A \sim \frac{\nabla F_s(\widehat{Q}_s)}{\|\nabla F_s(\widehat{Q}_s)\|_1}$$

and its output is modified to

$$F_s(\widehat{Q}_s) - \widehat{Q}_s^\mathsf{T}\nabla F_s(\widehat{Q}_s) + (R + \gamma\widehat{v})\|\nabla F_s(\widehat{Q})\|_1.$$

In particular, `SmoothCruiser` can be used for more general regularization schemes, as long as the Bellman operators satisfy the assumptions above. One such example is presented in Appendix E.

## 6   Conclusion

We provided `SmoothCruiser`, an algorithm that estimates the value function of MDPs and discounted games defined through smooth approximations of the optimal Bellman operator, which is the case in entropy-regularized value functions. More generally, our algorithm can also be used when value functions are defined through *any* smooth Bellman operator with nonnegative gradients. We showed that our algorithm has a polynomial sample complexity of $\widetilde{\mathcal{O}}(1/\varepsilon^4)$, where $\varepsilon$ is the desired precision. This guarantee is problem independent and holds for *state spaces of arbitrary cardinality.*

One interesting interpretation of our results is that computing entropy-regularized value functions, which are commonly employed for reinforcement learning, can be seen as a smooth relaxation of a planning problem for which we can obtain a much better sample complexity in terms of the required precision $\varepsilon$. Unsurprisingly, when the regularization tends to zero, we recover the well-known non-polynomial bound $\mathcal{O}\big((1/\varepsilon)^{\log(1/\varepsilon)}\big)$ of Kearns et al. [1999]. Hence, an interesting direction for future work is to study adaptive regularization schemes in order to accelerate planning algorithms. Although `SmoothCruiser` makes large amount of recursive calls, which makes it impractical in most situations, we believe it might help us to understand how regularization speeds planning and inspire more practical algorithms. This might be possible by exploiting its similarities to Monte-Carlo tree search that we have outlined above.

**Acknowledgments** The research presented was supported by European CHIST-ERA project DELTA, French Ministry of Higher Education and Research, Nord-Pas-de-Calais Regional Council, Inria and Otto-von-Guericke-Universität Magdeburg associated-team north-European project Allocate, and French National Research Agency project BoB (grant n.ANR-16-CE23-0003), FMJH Program PGMO with the support of this program from Criteo.

## Footnotes

[3]This means that the policy is seen as a function of time, not the states. The open-loop setting is particularly adapted to environments with deterministic transitions.

[4] $\nabla F_s(Q)$ is the gradient of $F_s(Q)$ with respect to $Q$.

[5] It is a Boltzmann distribution with temperature $\lambda$.

[6]This is the case of the $\max$ and $\min$ functions.

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
