[Supplementary Material · smoothcruiser2019_20200106.pdf]

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

# A  Preliminaries

## A.1  General definition of value functions

We consider the general definition of value functions in Equation 4 and we assume that all the functions $F_s$ satisfy

1. $F_s$ is differentiable,
2. $\forall x \in \mathbb{R}^K, 0 < \|\nabla F_s(x)\|_1 \leq 1$,
3. (nonnegative gradient) $\forall x \in \mathbb{R}^K, \nabla F_s(x) \succeq 0$,
4. ($L$-smooth) There exists $L \geq 0$ such that for any $x_0, x \in \mathbb{R}^K$,

$$|F_s(x) - F_s(x_0) - (x - x_0)^\intercal \nabla F_s(x_0)| \leq L\|x - x_0\|_2^2,$$

which is the case for the functions $\mathrm{LogSumExp}_\lambda$ and $-\mathrm{LogSumExp}_{-\lambda}$ that we study in the present paper. In particular, the second requirement implies that $F_s$ is 1-Lipschitz,

$$\forall x, y \in \mathbb{R}^K, |F_s(x) - F_s(y)| \leq \|x - y\|_\infty.$$

For this more general definition, we modify the output of `sampleV` when $\varepsilon < \kappa$ to

$$\textbf{output} = F_s\!\left(\widehat{Q}_s\right) - (\widehat{Q}_s)^\intercal \nabla F_s\!\left(\widehat{Q}_s\right) + (R + \gamma\widehat{v})\|\nabla F_s\!\left(\widehat{Q}\right)\|_1$$

and the action sampled in `sampleV` is sampled according to

$$A \sim \frac{\nabla F_s\!\left(\widehat{Q}_s\right)}{\|\nabla F_s\!\left(\widehat{Q}_s\right)\|_1} \quad \text{instead of} \quad A \sim \nabla F_s\!\left(\widehat{Q}_s\right).$$

## A.2  Other definitions

The constant $M_\lambda$ is defined as

$$M_\lambda \triangleq \sup_{s \in \mathcal{S}} |F_s(0)|.$$

For any $c \in \mathbb{R}$, the function $\mathrm{clip}_c : \mathbb{R}^d \to \mathbb{R}^d$ is defined component-wise as

$$\mathbf{clip}_c(x)_i = \begin{cases} 0 & \text{if } x_i \leq 0, \\ x_i & \text{if } -c < x_i < c, \\ c & \text{if } x \geq c. \end{cases}$$

# B  Sample complexity

**Theorem 1.** *Let $n(\varepsilon, \delta')$ be the number of oracle calls before `SmoothCruiser` terminates. For any state $s \in \mathcal{S}$ and $\varepsilon, \delta' > 0$,*

$$n(\varepsilon, \delta') \leq \frac{c_1}{\varepsilon^4} \log\!\left(\frac{c_2}{\delta'}\right) \left[c_3 \log\!\left(\frac{c_4}{\varepsilon}\right)\right]^{\log_2\left(c_5\left(\log\left(\frac{c_2}{\delta'}\right)\right)\right)} = \widetilde{\mathcal{O}}\!\left(\frac{1}{\varepsilon^4}\right),$$

*where $c_1, c_2, c_3, c_4$, and $c_5$ are constants that depend only on $K$, $L$, and $\gamma$.*

To bound the sample complexity, we make the following steps.

- Proposition 1 bounds the number of recursive calls of `sampleV` in the uniform sampling phase ($\varepsilon \geq \kappa$) and is similar to the results of Kearns et al. [1999].
- Lemma 1 bounds the number of recursive calls of `sampleV` when $\varepsilon < \kappa$.
- By noticing that the number of recursive calls of `sampleV` is equal to the number of oracle calls, we bound the sample complexity of `SmoothCruiser` in Theorem 1.

Let $n_{\texttt{sampleV}}(s, \varepsilon, \delta')$ be the total number of recursive calls to `sampleV` after an initial call with parameters $(s, \varepsilon)$, and including the initial call. Since this number does not depend on the state $s$, we denote it by $n_{\texttt{sampleV}}(\varepsilon, \delta')$.

**Proposition 1.** *Let $\varepsilon \geq \kappa$. For all $h \in \mathbb{N}$, $\forall \varepsilon$ such that $\frac{(1+M_\lambda)\sqrt{\gamma}^h}{1-\gamma} \leq \varepsilon \leq \frac{1+M_\lambda}{1-\gamma}$, we have*

$$n_{\texttt{sampleV}}(\varepsilon, \delta') \leq \gamma^{\frac{1}{2} H(\varepsilon)(H(\varepsilon)-1)} \left( \frac{2\alpha(\delta')}{\varepsilon^2} \right)^{H(\varepsilon)}$$

$$\leq \gamma^{\frac{1}{2} H(\kappa)(H(\kappa)-1)} \left( \frac{2\alpha(\delta')}{\kappa^2} \right)^{H(\kappa)}$$

*where*

$$H(\varepsilon) = \left\lceil 2 \log_\gamma \left( \frac{\varepsilon(1-\gamma)}{1+M_\lambda} \right) \right\rceil$$

*and*

$$\alpha(\delta') = \frac{18(1+M_\lambda)^2 K}{(1-\gamma)^4 (1-\sqrt{\gamma})^2} \log \left( \frac{2K}{\delta'} \right)$$

*Proof.* We want to prove that $n_{\texttt{sampleV}}(\varepsilon, \delta') \leq G(\varepsilon)$, where

$$G(\varepsilon) = \gamma^{\frac{1}{2} H(\varepsilon)(H(\varepsilon)-1)} \left( \frac{2\alpha(\delta')}{\varepsilon^2} \right)^{H(\varepsilon)}$$

We proceed by induction on $h$.

**Base case** Let $h = 0$. We have $\varepsilon = \frac{1+M_\lambda}{1-\gamma}$, which implies $n_{\texttt{sampleV}}(\varepsilon, \delta') = 1$ and $G(\varepsilon) = 1$ (since $H(\varepsilon) = 0$). Hence, the proposition is true for $h = 0$.

**Induction hypothesis** Assume true for $h$.

**Induction step** Let $\varepsilon \geq \frac{(1+M_\lambda)\sqrt{\gamma}^{h+1}}{1-\gamma}$. Since $\frac{\varepsilon}{\sqrt{\gamma}} \geq \frac{(1+M_\lambda)\sqrt{\gamma}^h}{1-\gamma}$, we use the induction hypothesis to obtain

$$n_{\texttt{sampleV}}(\varepsilon, \delta') = \underbrace{1}_{\text{current call}} + \underbrace{KN(\varepsilon) n_{\texttt{sampleV}}\left( \frac{\varepsilon}{\sqrt{\gamma}}, \delta' \right)}_{\text{calls in } \texttt{estimateQ}}$$

$$\leq \frac{2\alpha(\delta')}{\varepsilon^2} n_{\texttt{sampleV}}\left( \frac{\varepsilon}{\sqrt{\gamma}}, \delta' \right)$$

$$\leq \frac{2\alpha(\delta')}{\varepsilon^2} \gamma^{\frac{1}{2}(H(\varepsilon)-1)(H(\varepsilon)-2)} \left( \frac{\gamma 2\alpha(\delta')}{\varepsilon^2} \right)^{H(\varepsilon)-1}, \qquad \text{since } H\left( \frac{\varepsilon}{\sqrt{\gamma}} \right) = H(\varepsilon) - 1$$

$$= \gamma^{\frac{1}{2} H(\varepsilon)(H(\varepsilon)-1)} \left( \frac{2\alpha(\delta')}{\varepsilon^2} \right)^{H(\varepsilon)},$$

which completes the proof. $\qquad \square$

**Lemma 1.** *Let $\varepsilon \leq \kappa$. For all $h \in \mathbb{N}$, $\forall \varepsilon \geq \kappa \sqrt{\gamma}^h$, we have*

$$n_{\texttt{sampleV}}(\varepsilon, \delta') \leq \eta_1 \left[ \log_{\frac{1}{\gamma}} \left( \frac{\kappa/\gamma}{\varepsilon} \right) \right]^{\eta_2(\delta')} \frac{1}{\varepsilon^2}$$

*where*

$$\kappa = \frac{1 - \sqrt{\gamma}}{KL}$$

$$\eta_1 = \kappa^2 n_{\mathtt{sampleV}}(\kappa, \delta')$$

$$\eta_2(\delta') = \log_2\left(\frac{\gamma}{1 - \gamma}\frac{2\beta(\delta')}{\kappa}\right)$$

$$\beta(\delta') = \frac{18(1 + M_\lambda)^2 K^2 L}{(1 - \gamma)^4 (1 - \sqrt{\gamma})^3}\log\left(\frac{2K}{\delta'}\right)$$

*under the condition that*

$$\log_2\left(\frac{\gamma}{1 - \gamma}\frac{2\beta(\delta')}{\kappa}\right) \geq 0, \quad i.e., \quad \beta(\delta') \geq \frac{(1 - \gamma)(1 - \sqrt{\gamma})}{2\gamma KL} \tag{6}$$

*which is satisfied by choosing $\delta'$ small enough.*

*Proof.* First, let us define some auxiliary quantities,

$$B_1(\varepsilon) \triangleq \left[\log_{\frac{1}{\gamma}}\left(\frac{\kappa/\gamma}{\varepsilon}\right)\right]^{\eta_2(\delta')}, \tag{7}$$

$$B_2(\varepsilon) \triangleq \frac{\eta_1}{\varepsilon^2} \quad \text{and} \tag{8}$$

$$B(\varepsilon) \triangleq B_1(\varepsilon) B_2(\varepsilon) \tag{9}$$

We want to prove that $n_{\mathtt{sampleV}}(\varepsilon, \delta') \leq B(\varepsilon)$ and we proceed by induction on $h$.

**Base case** For $h = 0$, we have $\varepsilon \geq \kappa$ and, by assumption, $\varepsilon \leq \kappa$. Therefore, $\varepsilon = \kappa$. It can be easily verified that $B(\kappa) = n_{\mathtt{sampleV}}(\kappa, \delta')$, hence the lemma is true for $h = 0$.

**Induction hypothesis** Assume that the lemma is true for $h$.

**Induction step** Let $\varepsilon \geq \kappa\sqrt{\gamma}^{h+1}$. We have that

$$n_{\mathtt{sampleV}}(\varepsilon, \delta') = \underbrace{1}_{\text{current call}} + \underbrace{n_{\mathtt{sampleV}}\left(\frac{\varepsilon}{\sqrt{\gamma}}, \delta'\right)}_{\text{call in line 11 of } \mathtt{sampleV}} + \underbrace{KN(\sqrt{\kappa\varepsilon})n_{\mathtt{sampleV}}\left(\sqrt{\frac{\kappa\varepsilon}{\gamma}}, \delta'\right)}_{\text{calls in } \mathtt{estimateQ}}$$

$$= 1 + n_{\mathtt{sampleV}}\left(\frac{\varepsilon}{\sqrt{\gamma}}, \delta'\right) + \frac{\beta(\delta')}{\varepsilon} n_{\mathtt{sampleV}}\left(\sqrt{\frac{\kappa\varepsilon}{\gamma}}, \delta'\right)$$

$$\leq n_{\mathtt{sampleV}}\left(\frac{\varepsilon}{\sqrt{\gamma}}, \delta'\right) + \frac{2\beta(\delta')}{\varepsilon} n_{\mathtt{sampleV}}\left(\sqrt{\frac{\kappa\varepsilon}{\gamma}}, \delta'\right)$$

Since $\varepsilon \geq \kappa\sqrt{\gamma}^{h+1}$ and $\varepsilon \leq \kappa$, we have $\sqrt{\frac{\kappa\varepsilon}{\gamma}} \geq \frac{\varepsilon}{\sqrt{\gamma}} \geq \kappa\sqrt{\gamma}^h$. This allows us to use our induction hypothesis to get

$$n_{\mathtt{sampleV}}(\varepsilon, \delta') \leq B\left(\frac{\varepsilon}{\sqrt{\gamma}}\right) + \frac{2\beta(\delta')}{\varepsilon} B\left(\sqrt{\frac{\kappa\varepsilon}{\gamma}}\right).$$

We will need the equation bellow, which is easily verified as

$$\log\left(\frac{\kappa}{\sqrt{\frac{\kappa\varepsilon}{\gamma}}\gamma}\right) = \frac{1}{2}\log\left(\frac{\kappa/\gamma}{\varepsilon}\right) \tag{10}$$

We have that

$$\frac{B\left(\frac{\varepsilon}{\sqrt{\gamma}}\right)}{B(\varepsilon)} = \frac{B_1\left(\frac{\varepsilon}{\sqrt{\gamma}}\right)}{B_1(\varepsilon)}\frac{B_2\left(\frac{\varepsilon}{\sqrt{\gamma}}\right)}{B_2(\varepsilon)}$$

$$= \gamma \underbrace{\left[\frac{\log\left(\frac{\kappa/\gamma}{\varepsilon}\right) - \frac{1}{2}\log\frac{1}{\gamma}}{\log\left(\frac{\kappa/\gamma}{\varepsilon}\right)}\right]^{\eta_2(\delta')}}_{<1}$$

$$\leq \gamma,$$

where we used the assumption that $\eta_2(\delta') \geq 0$.

Also we get that

$$\frac{B\left(\sqrt{\frac{\kappa\varepsilon}{\gamma}}\right)}{B(\varepsilon)} = \frac{\varepsilon\gamma}{\kappa}\frac{B_1\left(\sqrt{\frac{\kappa\varepsilon}{\gamma}}\right)}{B_1(\varepsilon)}$$

$$= \frac{\varepsilon\gamma}{\kappa}\left[\frac{\log_{\frac{1}{\gamma}}\left(\frac{\kappa}{\sqrt{\frac{\kappa\varepsilon}{\gamma}}\gamma}\right)}{\log_{\frac{1}{\gamma}}\left(\frac{\kappa/\gamma}{\varepsilon}\right)}\right]^{\eta_2(\delta')}$$

$$= \frac{\varepsilon\gamma}{\kappa}\left[\frac{\frac{1}{2}\log_{\frac{1}{\gamma}}\left(\frac{\kappa/\gamma}{\varepsilon}\right)}{\log_{\frac{1}{\gamma}}\left(\frac{\kappa/\gamma}{\varepsilon}\right)}\right]^{\eta_2(\delta')}$$

$$= \frac{\varepsilon\gamma}{\kappa}\left(\frac{1}{2}\right)^{\eta_2(\delta')} = \frac{\varepsilon\gamma}{\kappa}\frac{(1-\gamma)}{\gamma}\frac{\kappa}{2\beta(\delta')} = \frac{(1-\gamma)\varepsilon}{2\beta(\delta')}$$

Finally, we obtain

$$n_{\texttt{sampleV}}(\varepsilon,\delta') \leq B\left(\frac{\varepsilon}{\sqrt{\gamma}}\right) + \frac{2\beta(\delta')}{\varepsilon}B\left(\sqrt{\frac{\kappa\varepsilon}{\gamma}}\right)$$

$$\leq \gamma B(\varepsilon) + \frac{2\beta(\delta')}{\varepsilon}\frac{(1-\gamma)\varepsilon}{2\beta(\delta')}B(\varepsilon)$$

$$= B(\varepsilon),$$

which proves the lemma. $\qquad\square$

Now we can prove Theorem 1, which is restated below.

**Theorem.** *Let $n(\varepsilon,\delta')$ be the number of calls to the generative model (oracle) before the algorithm terminates. For any state $s \in \mathcal{S}$ and $\varepsilon, \delta' > 0$,*

$$n(\varepsilon,\delta') \leq \frac{c_1}{\varepsilon^4}\log\left(\frac{c_2}{\delta'}\right)\left[c_3\log\left(\frac{c_4}{\varepsilon}\right)\right]^{\log_2\left(c_5\left(\log\left(\frac{c_2}{\delta'}\right)\right)\right)} = \widetilde{\mathcal{O}}\left(\frac{1}{\varepsilon^4}\right)$$

*where $c_1, c_2, c_3, c_4$ and $c_5$ are constants that depend only on $K$, $L$ and $\gamma$.*

*Proof.* First, notice that the number of calls to the generative model is smaller than the total number of calls to `sampleV`. `SmoothCruiser` makes one call to `estimateQ`, which makes $N(\varepsilon)$ calls to `sampleV`. If $\varepsilon \geq \kappa$, Proposition 1 shows that the sample complexity is bounded by a constant. Lemma 1 bounds the sample complexity for $\varepsilon \leq \kappa$, and we use it to bound $n(\varepsilon,\delta')$:

$$n(\varepsilon,\delta') = N(\varepsilon)n_{\texttt{sampleV}}(\varepsilon,\delta')$$

$$\leq N(\varepsilon)\eta_1\left[\log_{\frac{1}{\gamma}}\left(\frac{\kappa/\gamma}{\varepsilon}\right)\right]^{\eta_2(\delta')}\frac{1}{\varepsilon^2}$$

$$\leq \frac{c_1}{\varepsilon^4}\log\left(\frac{c_2}{\delta'}\right)\left[c_3\log\left(\frac{c_4}{\varepsilon}\right)\right]^{\log_2\left(c_5\left(\log\left(\frac{c_2}{\delta'}\right)\right)\right)} = \widetilde{\mathcal{O}}\left(\frac{1}{\varepsilon^4}\right)$$

by using the definition of $N(\varepsilon)$ for $\varepsilon \leq \kappa$ and the definition of $\eta_2(\delta')$ in Lemma 1.

The constants are given by:

- $c_1 = \frac{18(1+M_\lambda)^2 n_{\mathtt{sampleV}}(\kappa,\delta')}{K^2 L^2 (1-\gamma)^4}$;

- $c_2 = 2K$;

- $c_3 = [\log(1/\gamma)]^{-1}$;

- $c_4 = (1 - \sqrt{\gamma})/(\gamma K L)$;

- $c_5 = \frac{36(1+M_\lambda)^2 \gamma K^3 L^2}{(1-\gamma)^5 (1-\sqrt{\gamma})^4}$.

$\square$

## C  Consistency

**Theorem 2.** *For any $s \in \mathcal{S}$, $\varepsilon > 0$, and $\delta > 0$, there exists a $\delta'$ that depends on $\varepsilon$ and $\delta$ such that the output $\widehat{V}(s)$ of* $\mathtt{SmoothCruiser}(s, \varepsilon, \delta')$ *satisfies*

$$\mathbb{P}\Big[\big|\widehat{V}(s) - V(s)\big| > \varepsilon\Big] \leq \delta.$$

*and such that $n(\varepsilon, \delta') = \mathcal{O}\big(1/\varepsilon^{4+c}\big)$ for any $c > 0$.*

To prove that our algorithm outputs a good estimate of the value function with high probability, we proceed as follows:

- In Lemma 2, we prove that the output of $\mathtt{sampleV}$, conditioned on an event $\mathcal{A}$, is a low-bias estimate of the true value function, and that $\mathcal{A}$ happens with high probability;
- Given Lemma 2, the proof of Theorem 2 is straightforward.

Throughout the proof, we will make distinctions between two cases:

- **Case 1:** $\kappa \leq \varepsilon < \frac{1+M_\lambda}{1-\gamma}$
- **Case 2:** $\varepsilon < \kappa$

### C.1  Definitions

We define the function $\zeta(\varepsilon)$ as

$$\zeta(\varepsilon) = \begin{cases} \varepsilon, & \text{if } \kappa \leq \varepsilon < \frac{1+M_\lambda}{1-\gamma}, \\ \sqrt{\kappa\varepsilon}, & \text{if } \varepsilon < \kappa, \\ \infty, & \text{otherwise.} \end{cases} \quad (11)$$

Define $\mathbf{params}(s, \varepsilon)$ as the (random) set of parameters used to call $\mathtt{sampleV}$ after a call to $\mathtt{sampleV}(s, \varepsilon)$, that is

$$\mathbf{params}(s, \varepsilon) = \left\{ \left( Z_{s,a}^{(k)}, \frac{\zeta(\varepsilon)}{\sqrt{\gamma}} \right) \text{ for } k = 1, \ldots, N(\varepsilon); a \in \mathcal{A} \right\} \quad (12)$$

in case 1 and

$$\mathbf{params}(s, \varepsilon) = \left\{ \left( Z_{s,a}^{(k)}, \frac{\zeta(\varepsilon)}{\sqrt{\gamma}} \right) \text{ for } k = 1, \ldots, N(\varepsilon); a \in \mathcal{A} \right\} \bigcup \left\{ \left( Z_{s,A}, \frac{\varepsilon}{\sqrt{\gamma}} \right) \right\} \quad (13)$$

in case 2, where $Z_{s,a}^{(k)}$ are the next states sampled in `estimateQ` and $Z_{s,A}$ is the next state sampled `sampleV`$(s, \varepsilon)$.

A call to `sampleV`$(s, \varepsilon)$ makes one call to `estimateQ`. Denote the output of this call to `estimateQ` by $\widehat{Q}_s^\varepsilon$. We define the event $\mathcal{A}(s, \varepsilon)$ as follows:

$$\mathcal{A}(s, \varepsilon) = \begin{cases} \left\{ \|\widehat{Q}_s^\varepsilon - Q_s\|_\infty \le \zeta(\varepsilon) \right\} \bigcap \mathcal{B}(s, \varepsilon), & \text{if } 0 < \varepsilon < \frac{1+M_\lambda}{1-\gamma}, \\ \Omega, & \text{if } \varepsilon \ge \frac{1+M_\lambda}{1-\gamma}. \end{cases} \tag{14}$$

where $\Omega$ is the whole sample space and

$$\mathcal{B}(s, \varepsilon) = \bigcap_{(z,e) \in \mathbf{params}(s,\varepsilon)} \mathcal{A}(z, e) \tag{15}$$

Define $C_\gamma$ as:

$$C_\gamma = \frac{3(1 + M_\lambda)}{(1-\gamma)^2} \tag{16}$$

## C.2   Proofs

**Lemma 2.** *Let* $\widehat{V}_\varepsilon(s) = $ `sampleV`$(s, \varepsilon)$. *For all* $h \in \mathbb{N}, s \in \mathcal{S}, \varepsilon \ge \frac{(1+M_\lambda)\sqrt{\gamma}^h}{1-\gamma}$, *we have:*

$$\text{(i)} \quad \left| \mathbb{E}\left[ \widehat{V}_\varepsilon(s) \Big| \mathcal{A}(s, \varepsilon) \right] - V(s) \right| \le \varepsilon, \text{ and}$$

$$\text{(ii)} \quad \mathbb{P}\left[ |\widehat{V}_\varepsilon(s)| \le C_\gamma \Big| \mathcal{A}(s, \varepsilon) \right] = 1$$

$$\text{(iii)} \quad \mathbb{P}[\mathcal{A}(s, \varepsilon)] \ge 1 - \delta' n_{\texttt{sampleV}}(\varepsilon, \delta')$$

*where*

$$n_{\texttt{sampleV}}(\varepsilon, \delta') = 1 + \sum_{(z,e) \in \mathbf{params}(s,\varepsilon)} n_{\texttt{sampleV}}(e, \delta') \tag{17}$$

*is the total number of recursive calls to* `sampleV` *after an initial call with parameters* $(s, \varepsilon)$.

*Proof.* We proceed by induction over $h$.

**(1) Base case.**   If $h = 0$, $\varepsilon \ge \frac{1+M_\lambda}{1-\gamma}$ and $\mathcal{A}(s, \varepsilon) = \Omega$. The output is then $\widehat{V}_\varepsilon(s) = 0$. Point **(i)** is verified by using the fact that $|V(s)| \le \frac{1+M_\lambda}{1-\gamma} \le \varepsilon$; points **(ii)** and **(iii)** are trivially verified.

**(2) Induction hypothesis.**   Assume that **(i)**, **(ii)** and **(iii)** are true for $h$.

**(3) Induction step.**   Let $\varepsilon \ge \frac{(1+M_\lambda)\sqrt{\gamma}^{h+1}}{1-\gamma}$. This implies that $\varepsilon/\sqrt{\gamma}$ and $\zeta(\varepsilon)/\sqrt{\gamma}$ are both greater than $\frac{(1+M_\lambda)\sqrt{\gamma}^h}{1-\gamma}$, which will allow us to use our induction hypothesis.

We start by proving **(iii)**.

Let $\widehat{Q}_s^\varepsilon = $ `estimateQ`$(s, \zeta(\varepsilon))$. Let the reward $R_{s,a}^{(k)}$ and state $Z_{s,a}^{(k)}$ be the random variables associated to the $k$-th call to the generative model used to compute $\widehat{Q}_s$ in `estimateQ`, for $k \in \{1, \cdots, N(\varepsilon)\}$. Let

$$q_s^k(a) := R_{s,a}^{(k)} + \gamma \texttt{sampleV}\left( Z_{s,a}^{(k)}, \zeta(\varepsilon)/\sqrt{\gamma} \right) \tag{18}$$

and let

$$\overline{Q}_s^\varepsilon(a) = \frac{1}{N(\varepsilon)} \sum_{k=1}^{N(\varepsilon)} q_s^k(a) \tag{19}$$

so that:

$$\widehat{Q}_s^\varepsilon = \mathbf{clip}_{(1+M_\lambda)(1-\gamma)^{-1}}\left(\overline{Q}_s^\varepsilon(a)\right) \tag{20}$$

Using Fact 2, we have:

$$|\widehat{Q}_s^\varepsilon(a) - Q_s(a)| \leq |\overline{Q}_s^\varepsilon(a) - Q_s(a)| \tag{21}$$

$$\leq \underbrace{|\overline{Q}_s^\varepsilon(a) - \mathbb{E}\left[\overline{Q}_s^\varepsilon(a)|\mathcal{B}(s,\varepsilon)\right]|}_{\textbf{(I)}} + \underbrace{|\mathbb{E}\left[\overline{Q}_s^\varepsilon(a)|\mathcal{B}(s,\varepsilon)\right] - Q_s(a)|}_{\textbf{(II)}} \tag{22}$$

We'd like to use Hoeffding's inequality to bound **(I)** in probability. For that, we need to verify that the random variables $\{q_s^k(a)\}_{k=1}^{N(\varepsilon)}$ are bounded and independent conditionally on $\mathcal{B}(s,\varepsilon)$.

**Boundedness.** By induction hypothesis **(ii)** In the event $\mathcal{B}(s,\varepsilon)$, the random variables $\mathtt{sampleV}\left(Z_{s,a}^{(k)}, \zeta(\varepsilon)/\sqrt{\gamma}\right)$, for all $k$, are bounded by $C_\gamma$. Using the fact that the rewards are in $[0,1]$ and that $C_\gamma \geq 1/(1-\gamma)$, we obtain $q_s^k(a)$ is also bounded by $C_\gamma$.

**Independence.** Let $E_k = \mathcal{A}\left(Z_{s,a}^k, \zeta(\varepsilon)/\sqrt{\gamma}\right)$. For any $t \in \mathbb{R}^{N(\varepsilon)}$, the characteristic function of $\{q_s^k(a)\}_{k=1}^{N(\varepsilon)}$ conditionally on $\mathcal{B}(s,\varepsilon)$ is given by

$$\mathbb{E}\left[\exp\left(\mathrm{i}\sum_k t_k q_s^k(a)\right)\Big|\mathcal{B}(s,\varepsilon)\right] \overset{\textbf{(a)}}{=} \mathbb{E}\left[\exp\left(\mathrm{i}\sum_k t_k q_s^k(a)\right)\Big|\bigcap_k E_k\right]$$

$$= \frac{\mathbb{E}\left[\exp\left(\mathrm{i}\sum_k t_k q_s^k(a)\right)\prod_k \mathbb{I}_{\{E_k\}}\right]}{\mathbb{E}\left[\prod_k \mathbb{I}_{\{E_k\}}\right]}$$

$$= \frac{\mathbb{E}\left[\prod_k \exp\left(\mathrm{i}t_k q_s^k(a)\right)\mathbb{I}_{\{E_k\}}\right]}{\mathbb{E}\left[\prod_k \mathbb{I}_{\{E_k\}}\right]}$$

$$\overset{\textbf{(b)}}{=} \frac{\prod_k \mathbb{E}\left[\exp\left(\mathrm{i}t_k q_s^k(a)\right)\mathbb{I}_{\{E_k\}}\right]}{\prod_k \mathbb{E}\left[\mathbb{I}_{\{E_k\}}\right]}$$

$$= \prod_k \mathbb{E}\left[\exp\left(\mathrm{i}t_k q_s^k(a)\right)\Big|E_k\right]$$

$$\overset{\textbf{(c)}}{=} \prod_k \mathbb{E}\left[\exp\left(\mathrm{i}t_k q_s^k(a)\right)\Big|\mathcal{B}(s,\varepsilon)\right]$$

which is justified by

(a) Definition of $\mathcal{B}(s,\varepsilon)$ and the fact that $\{q_s^k(a)\}_{k=1}^{N(\varepsilon)}$ are independent of $\mathcal{A}\left(Z_{s,A}, \frac{\varepsilon}{\sqrt{\gamma}}\right)$;

(b) The random variables $\{q_s^k(a)\}_{k=1}^{N(\varepsilon)}$ are independent and the events $\{E_k\}_{i=1}^{N(\varepsilon)}$ are also independent;

(c) The random variable $q_s^k(a)$ is independent of every $E_j$ for $j \neq k$.

Since the characteristic function of $\{q_s^k(a)\}_{k=1}^{N(\varepsilon)}$ is the product of their characteristic functions, these random variables are independent given $\mathcal{B}(s,\varepsilon)$.

Now we can use Hoeffding's inequality:

$$\mathbb{P}\Big[|\overline{Q}_s^\varepsilon(a) - \mathbb{E}\big[\overline{Q}_s^\varepsilon(a)\big|\mathcal{B}(s,\varepsilon)\big]| \geq (1 - \sqrt{\gamma})\zeta(\varepsilon)\Big|\mathcal{B}(s,\varepsilon)\Big]$$

$$= \mathbb{P}\left[|\frac{1}{N(\varepsilon)}\sum_{k=1}^{N(\varepsilon)} q_s^k(a) - \mathbb{E}\big[q_s^k(a)\big|\mathcal{B}(s,\varepsilon)\big]| \geq (1 - \sqrt{\gamma})\zeta(\varepsilon)\Big|\mathcal{B}(s,\varepsilon)\right]$$

$$\leq 2\exp\left(-\frac{N(\varepsilon)(1 - \sqrt{\gamma})^2\zeta(\varepsilon)^2}{2C_\gamma^2}\right)$$

$$\leq \frac{\delta'}{K}$$

And **(II)** is bounded by using the induction hypothesis **(i)**:

$$|\mathbb{E}\big[q_s^k(a)\big|\mathcal{B}(s,\varepsilon)\big] - Q_s(a)|$$

$$\overset{\text{(a)}}{=} \gamma|\mathbb{E}\big[\mathtt{sampleV}\big(Z_{s,a}^{(k)}, \tfrac{\zeta(\varepsilon)}{\sqrt{\gamma}}\big)\big|\mathcal{B}(s,\varepsilon)\big] - \mathbb{E}\big[V(Z_{s,a}^{(k)})\big|\mathcal{B}(s,\varepsilon)\big]|$$

$$\overset{\text{(b)}}{=} \gamma|\mathbb{E}\big[\mathtt{sampleV}\big(Z_{s,a}^{(k)}, \tfrac{\zeta(\varepsilon)}{\sqrt{\gamma}}\big)\big|\mathcal{A}\big(Z_{s,a}^{(k)}, \tfrac{\zeta(\varepsilon)}{\sqrt{\gamma}}\big)\big] - \mathbb{E}\big[V(Z_{s,a}^{(k)})\big|\mathcal{A}\big(Z_{s,a}^{(k)}, \tfrac{\zeta(\varepsilon)}{\sqrt{\gamma}}\big)\big]|$$

$$\overset{\text{(c)}}{=} \gamma|\mathbb{E}\big[\mathbb{E}\big[\mathtt{sampleV}\big(Z_{s,a}^{(k)}, \tfrac{\zeta(\varepsilon)}{\sqrt{\gamma}}\big)\big|Z_{s,a}^{(k)}, \mathcal{A}\big(Z_{s,a}^{(k)}, \tfrac{\zeta(\varepsilon)}{\sqrt{\gamma}}\big)\big] - V(Z_{s,a}^{(k)})\big|\mathcal{A}\big(Z_{s,a}^{(k)}, \tfrac{\zeta(\varepsilon)}{\sqrt{\gamma}}\big)\big]|$$

$$\overset{\text{(d)}}{\leq} \gamma\frac{\zeta(\varepsilon)}{\sqrt{\gamma}}$$

$$= \sqrt{\gamma}\zeta(\varepsilon)$$

which is justified by the following:

(a) $\mathbb{E}\big[R_{s,a}^{(k)}\big|\mathcal{B}(s,\varepsilon)\big] = \mathbb{E}\big[R_{s,a}^{(k)}\big]$, since the reward depends only on $s, a$;

(b) The term $\big(Z_{s,a}^{(k)}, \tfrac{\zeta(\varepsilon)}{\sqrt{\gamma}}\big)$ depends on $\mathcal{B}(s,\varepsilon)$ only through $\mathcal{A}\big(Z_{s,a}^{(k)}, \tfrac{\zeta(\varepsilon)}{\sqrt{\gamma}}\big)$;

(c) Law of total expectation;

(d) Consequence of induction hypothesis **(i)**.

Putting together the bounds for **(I)** and **(II)** and doing an union bound over all actions, we obtain:

$$\mathbb{P}\Big[\|\widehat{Q}_s^\varepsilon - Q_s\|_\infty \geq \zeta(\varepsilon)\Big|\mathcal{B}(s,\varepsilon)\Big] \leq \delta'$$

We can now give a lower bound to the probability of the event $\mathcal{A}(s,\varepsilon)$. Let

$$\mathcal{E} = \Big\{\|\widehat{Q}_s^\varepsilon - Q_s\|_\infty < \zeta(\varepsilon)\Big\} \tag{23}$$

We have:

$$\mathbb{P}[\mathcal{A}(s,\varepsilon)] \geq \mathbb{P}[\mathcal{E} \cap \mathcal{B}(s,\varepsilon)]$$
$$= \mathbb{P}\left[\mathcal{E}\middle|\mathcal{B}(s,\varepsilon)\right]\mathbb{P}[\mathcal{B}(s,\varepsilon)]$$
$$= \left(1 - \mathbb{P}\left[\mathcal{E}^{\complement}\middle|\mathcal{B}(s,\varepsilon)\right]\right)\mathbb{P}[\mathcal{B}(s,\varepsilon)]$$
$$\geq \mathbb{P}[\mathcal{B}(s,\varepsilon)] - \delta'$$
$$\geq 1 - \delta' n_{\texttt{sampleV}}(\varepsilon,\delta')$$

since

$$\mathbb{P}[\mathcal{B}(s,\varepsilon)] = 1 - \mathbb{P}\left[\mathcal{B}(s,\varepsilon)^{\complement}\right]$$
$$= 1 - \mathbb{P}\left[\bigcup_{(z,e)\in\mathbf{params}(s,\varepsilon)} \mathcal{A}(z,e)^{\complement}\right]$$
$$\geq 1 - \sum_{(z,e)\in\mathbf{params}(s,\varepsilon)} \mathbb{P}\left[\mathcal{A}(z,e)^{\complement}\right]$$
$$\geq 1 - \delta' \sum_{(z,e)\in\mathbf{params}(s,\varepsilon)} n_{\texttt{sampleV}}(e,\delta') \quad \text{by induction hypothesis (iii)}$$
$$= 1 - \delta'(n_{\texttt{sampleV}}(\varepsilon,\delta') - 1)$$

This proves **(iii)**. Now, let's prove **(i)**.

For any event $\mathcal{E}$, we write

$$\mathbb{E}_{\mathcal{E}}[\cdot] = \mathbb{E}\left[\cdot\middle|\mathcal{E}\right]$$

**Case 1.**    We start with case 1, $\kappa \leq \varepsilon < \frac{1+M_\lambda}{1-\gamma}$, where $\zeta(\varepsilon) = \varepsilon$ and

$$\widehat{V}_\varepsilon(s) = F_s(\widehat{Q}_s^\varepsilon) \tag{24}$$

We have:

$$|\mathbb{E}_{\mathcal{A}(s,\varepsilon)}\left[\widehat{V}_\varepsilon(s)\right] - V(s)| = |\mathbb{E}_{\mathcal{A}(s,\varepsilon)}\left[F_s(\widehat{Q}_s^\varepsilon) - F_s(Q_s)\right]|$$
$$\leq \mathbb{E}_{\mathcal{A}(s,\varepsilon)}\left[|F_s(\widehat{Q}_s^\varepsilon) - F_s(Q_s)|\right]$$
$$\leq \mathbb{E}_{\mathcal{A}(s,\varepsilon)}\left[\|\widehat{Q}_s^\varepsilon(a) - Q_s(a)\|_\infty\right]$$
$$\leq \zeta(\varepsilon) = \varepsilon$$

and **(i)** is verified for case 1.

**Case 2.**    Consider now the case 2, $\varepsilon < \kappa$, where $\zeta(\varepsilon) = \sqrt{\kappa\varepsilon}$.

Let $A$ be the action following the distribution $\frac{\nabla F_s(\widehat{Q}_s^\varepsilon)}{\|\nabla F_s(\widehat{Q}_s^\varepsilon)\|_1}$, and let the reward $R_{s,A}$ and the state $Z_{s,A}$ be the random variables associated to the call to the generative model with parameters $(s,A)$. Let $\widehat{v} = \texttt{sampleV}(Z_{s,A}, \varepsilon/\sqrt{\gamma})$. The output in this case is given by

$$\widehat{V}_\varepsilon(s) = F_s\left(\widehat{Q}_s^\varepsilon\right) - (\widehat{Q}_s^\varepsilon)^\intercal \nabla F_s\left(\widehat{Q}_s^\varepsilon\right) + (R + \gamma\widehat{v})\|\nabla F_s\left(\widehat{Q}_s^\varepsilon\right)\|_1 \tag{25}$$

Let

$$Q_s(A) = \mathbb{E}_{\mathcal{A}(s,\varepsilon)}\left[R_{s,A} + \gamma V(Z_{s,A})|A, \widehat{Q}_s^\varepsilon\right]$$
$$= \mathbb{E}_{\mathcal{A}(s,\varepsilon)}[R_{s,A} + \gamma V(Z_{s,A})|A]$$

and let

$$\widetilde{V}(s) = \mathbb{E}_{\mathcal{A}(s,\varepsilon)}\left[F_s\left(\widehat{Q}_s^\varepsilon\right) - (\widehat{Q}_s^\varepsilon)^\top \nabla F_s\left(\widehat{Q}_s^\varepsilon\right) + Q_s(A)\|\nabla F_s\left(\widehat{Q}_s^\varepsilon\right)\|_1\right] \qquad (26)$$

We have

$$\left|\mathbb{E}_{\mathcal{A}(s,\varepsilon)}\left[\widehat{V}_\varepsilon(s)\right] - \widetilde{V}(s)\right|$$

$$\overset{(a)}{=} \gamma\left|\mathbb{E}_{\mathcal{A}(s,\varepsilon)}\left[\mathbb{E}_{\mathcal{A}(s,\varepsilon)}\left[\mathtt{sampleV}\left(Z_{s,A}, \frac{\varepsilon}{\sqrt{\gamma}}\right) - V(Z_{s,A})\Big|A, \widehat{Q}_s^\varepsilon, Z_{s,A}\right]\|\nabla F_s\left(\widehat{Q}_s^\varepsilon\right)\|_1\right]\right|$$

$$\overset{(b)}{=} \gamma\left|\mathbb{E}_{\mathcal{A}(s,\varepsilon)}\left[\left(\mathbb{E}_{\mathcal{A}(s,\varepsilon)}\left[\mathtt{sampleV}\left(Z_{s,A}, \frac{\varepsilon}{\sqrt{\gamma}}\right)\Big|A, \widehat{Q}_s^\varepsilon, Z_{s,A}\right] - V(Z_{s,A})\right)\right]\|\nabla F_s\left(\widehat{Q}_s^\varepsilon\right)\|_1\right|$$

$$\overset{(c)}{\le} \gamma\mathbb{E}_{\mathcal{A}(s,\varepsilon)}\left[\left|\mathbb{E}_{\mathcal{A}(s,\varepsilon)}\left[\mathtt{sampleV}\left(Z_{s,A}, \frac{\varepsilon}{\sqrt{\gamma}}\right)\Big|A, \widehat{Q}_s^\varepsilon, Z_{s,A}\right] - V(Z_{s,A})\right|\right]$$

$$\overset{(d)}{=} \gamma\mathbb{E}_{\mathcal{A}(s,\varepsilon)}\left[\left|\mathbb{E}_{\mathcal{A}(Z_{s,A},\varepsilon/\sqrt{\gamma})}\left[\mathtt{sampleV}\left(Z_{s,A}, \frac{\varepsilon}{\sqrt{\gamma}}\right)\Big|Z_{s,A}\right] - V(Z_{s,A})\right|\right]$$

$$\overset{(e)}{\le} \gamma\frac{\varepsilon}{\sqrt{\gamma}} = \sqrt{\gamma}\varepsilon$$

which is justified by the following points:

(a) The reward depend only on $s, a$ and law of total expectation;

(b) $V(Z_{s,A})$ is a function of $Z_{s,A}$ and no other random variable;

(c) Jensen's inequality and the fact that $\|\nabla F_s\left(\widehat{Q}_s^\varepsilon\right)\|_1 \le 1$;

(d) Given $Z_{s,A}$, the term $\mathtt{sampleV}\left(Z_{s,A}, \frac{\varepsilon}{\sqrt{\gamma}}\right)$ depends on $\mathcal{A}(s,\varepsilon)$ only through $\mathcal{A}(Z_{s,A}, \varepsilon/\sqrt{\gamma})$;

(e) Induction hypothesis **(i)**.

Now, $\mathbb{E}_{\mathcal{A}(s,\varepsilon)}\left[Q_s(A)\|\nabla F_s\left(\widehat{Q}_s^\varepsilon\right)\|_1\right]$ can be written as

$$\mathbb{E}_{\mathcal{A}(s,\varepsilon)}\left[Q_s(A)\|\nabla F_s\left(\widehat{Q}_s^\varepsilon\right)\|_1\right]$$
$$= \mathbb{E}_{\mathcal{A}(s,\varepsilon)}\left[\mathbb{E}_{\mathcal{A}(s,\varepsilon)}\left[Q_s(A)\Big|\widehat{Q}_s^\varepsilon\right]\|\nabla F_s\left(\widehat{Q}_s^\varepsilon\right)\|_1\right]$$
$$= \mathbb{E}_{\mathcal{A}(s,\varepsilon)}\left[Q_s^\top \nabla F_s\left(\widehat{Q}_s^\varepsilon\right)\right]$$

so that $\widetilde{V}(s)$ is given by

$$\widetilde{V}(s) = \mathbb{E}_{\mathcal{A}(s,\varepsilon)}\left[F_s\left(\widehat{Q}_s^\varepsilon\right) + (Q_s - \widehat{Q}_s^\varepsilon)^\top \nabla F_s\left(\widehat{Q}_s^\varepsilon\right)\right] \qquad (27)$$

Finally, we bound the difference between $\widetilde{V}(s)$ and $V(s)$:

$$\left| \widetilde{V}(s) - V(s) \right| \leq \mathbb{E}_{\mathcal{A}(s,\varepsilon)} \left[ \left| F_s\left( \widehat{Q}_s^\varepsilon \right) + (Q_s - \widehat{Q}_s^\varepsilon)^\intercal \nabla F_s\left( \widehat{Q}_s^\varepsilon \right) - V(s) \right| \right]$$

$$\leq L \mathbb{E}_{\mathcal{A}(s,\varepsilon)} \left[ \| Q_s - \widehat{Q}_s^\varepsilon \|_2^2 \right]$$

$$\overset{(\mathbf{a})}{\leq} KL \mathbb{E}_{\mathcal{A}(s,\varepsilon)} \left[ \| Q_s - \widehat{Q}_s^\varepsilon \|_\infty^2 \right]$$

$$\leq KL \zeta(\varepsilon)^2$$

$$= KL\kappa\varepsilon$$

$$= (1 - \sqrt{\gamma})\varepsilon$$

by using the fact that we are on $\mathcal{A}(s,\varepsilon)$ and (a) uses the fact that for all $x \in \mathbb{R}^K$, $\|x\|_2^2 \leq K\|x\|_\infty^2$.
We can now prove **(i)** for case 2:

$$\left| \mathbb{E}_{\mathcal{A}(s,\varepsilon)} \left[ \widehat{V}_\varepsilon(s) \right] - V(s) \right| \leq \left| \mathbb{E}_{\mathcal{A}(s,\varepsilon)} \left[ \widehat{V}_\varepsilon(s) \right] - \widetilde{V}(s) \right| + |\widetilde{V}(s) - V(s)| \tag{28}$$

$$\leq \sqrt{\gamma}\varepsilon + (1 - \sqrt{\gamma})\varepsilon = \varepsilon \tag{29}$$

Finally, let's prove **(ii)**.

**Case 1.** In this case, $\widehat{V}_\varepsilon(s) = F_s(\widehat{Q}_s^\varepsilon)$ with $\|\widehat{Q}_s^\varepsilon\|_\infty \leq (1 + M_\lambda)/(1 - \gamma)$, since each component of $\widehat{Q}_s^\varepsilon$ is clipped and lie in the interval $\left[ 0, \frac{1+M_\lambda}{1-\gamma} \right]$. The assumptions on $F_s$ imply that $|\widehat{V}_\varepsilon(s)| \leq \frac{1+M_\lambda}{1-\gamma} \leq C_\gamma$.

**Case 2.** In this case, we have:

$$\left| \widehat{V}_\varepsilon(s) \right| \leq \left| F_s\left( \widehat{Q}_s^\varepsilon \right) - (\widehat{Q}_s^\varepsilon)^\intercal \nabla F_s\left( \widehat{Q}_s^\varepsilon \right) \right| + \| R + \gamma \widehat{v} \| \| \nabla F_s\left( \widehat{Q}_s^\varepsilon \right) \|_1$$

$$\leq 2\|\widehat{Q}_s^\varepsilon\|_\infty + M_\lambda + 1 + \gamma C_\gamma$$

$$\leq \frac{2(1 + M_\lambda)}{1 - \gamma} + M_\lambda + 1 + \gamma C_\gamma$$

$$\leq C_\gamma$$

since $|\widehat{v}| \leq C_\gamma$ by induction hypothesis **(ii)**.
This proves **(ii)** for case 2:

$$\mathbb{P}\left[ |\widehat{V}(s)| \leq C_\gamma \Big| \mathcal{A}(s,\varepsilon) \right] = 1 \tag{30}$$

$\square$

Now, we can prove Theorem 2, which is restated as follows:

**Theorem.** *Let $\widehat{V}(s)$ be the output of* SmoothCruiser$(s, \varepsilon, \delta')$. *For any state $s \in \mathcal{S}$ and $\varepsilon, \delta' > 0$,*

$$\mathbb{P}\left[ |\widehat{V}(s) - V(s)| > \varepsilon \right] \leq \delta' n(\varepsilon, \delta').$$

*Proof.* Let $\widehat{Q}_s = $ estimateQ$(s, \varepsilon)$. We have $\widehat{V}(s) = F_s(\widehat{Q}_s)$. As in the proof of Lemma 2, let the reward $R_{s,a}^{(k)}$ and state $Z_{s,a}^{(k)}$ be the random variables associated to the $k$-th call to the generative model used to compute $\widehat{Q}_s(a)$ in estimateQ, for $k \in \{1, \cdots, N(\varepsilon)\}$.
We have:

$$\widehat{Q}_s(a) = \frac{1}{N(\varepsilon)} \sum_{k=1}^{N(\varepsilon)} R_{s,a}^{(k)} + \gamma \, \mathtt{sampleV}\Big( Z_{s,a}^{(k)}, \varepsilon/\sqrt{\gamma} \Big) \tag{31}$$

Consider the event $\mathcal{E}$ defined by:

$$\mathcal{E} = \bigcap_{k=1}^{N(\varepsilon)} \mathcal{A}\Big( Z_{s,a}^{(k)}, \frac{\varepsilon}{\sqrt{\gamma}} \Big) \tag{32}$$

By the same arguments as in the proof of Lemma 2, we have:

- In $\mathcal{E}$, we have $\|\widehat{Q}_s - Q_s\|_\infty \le \varepsilon$;
- $\mathbb{P}[\mathcal{E}] \ge 1 - \delta' N(\varepsilon) n_{\mathtt{sampleV}}(\varepsilon, \delta') = 1 - \delta' n(\varepsilon, \delta')$.

This implies the result, since $|\widehat{V}(s) - V(s)| \le \|\widehat{Q}_s - Q_s\|_\infty$.

Now, for every $\varepsilon > 0$ and every $\delta > 0$, we need to be able to find a value of $\delta'$ such that $\delta' n(\varepsilon, \delta') \le \delta$. That is, given $\varepsilon$ and $\delta$, we need to find $\delta'$ such that

$$\delta' \frac{c_1}{\varepsilon^4} \log\Big( \frac{c_2}{\delta'} \Big) \Big[ c_3 \log\Big( \frac{c_4}{\varepsilon} \Big) \Big]^{\log_2\big( c_5 \big( \log\big( \frac{c_2}{\delta'} \big) \big) \big)} \le \delta. \tag{33}$$

Such value exists, since the term on the LHS tends to 0 as $\delta' \to 0$, and it depends on $\varepsilon$. We will show that this dependence is polynomial when $\varepsilon \to 0$.

Let $\delta' = \varepsilon^5$. There exists a value $\widetilde{\varepsilon}$ that depends on $\delta$ such that:

$$\forall \varepsilon \le \widetilde{\varepsilon}, \quad \varepsilon^5 \frac{c_1}{\varepsilon^4} \log\Big( \frac{c_2}{\varepsilon^5} \Big) \Big[ c_3 \log\Big( \frac{c_4}{\varepsilon} \Big) \Big]^{\log_2\big( c_5 \big( \log\big( \frac{c_2}{\varepsilon^5} \big) \big) \big)} \le \delta. \tag{34}$$

since the term on the LHS tends to 0 as $\varepsilon \to 0$, as a consequence of Proposition 2.

Putting it all together, we can choose $\delta'$ as folllows:

$$\delta' = \begin{cases} \widetilde{\delta} \text{ such that } \widetilde{\delta} \frac{c_1}{\varepsilon^4} \log\Big( \frac{c_2}{\widetilde{\delta}} \Big) \Big[ c_3 \log\Big( \frac{c_4}{\varepsilon} \Big) \Big]^{\log_2\big( c_5 \big( \log\big( \frac{c_2}{\widetilde{\delta}} \big) \big) \big)} \le \delta, & \text{if } \varepsilon > \widetilde{\varepsilon}, \\ \varepsilon^5, & \text{if } \varepsilon \le \widetilde{\varepsilon} \end{cases} \tag{35}$$

which is $\mathcal{O}\big( \varepsilon^5 \big)$.

Proposition 3 implies that, for this choice of $\delta'$, the sample complexity is still of order $\mathcal{O}\big( 1/\varepsilon^{4+c} \big)$ for any $c > 0$.

$\square$

# D    Auxiliary results

**Fact 1.** *For all $s \in \mathcal{S}$ and all $x \in \mathbb{R}^K$, we have $F_s(x) \le \|x\|_\infty + \sup_s |F_s(0)|$.*

*Proof.* By the assumptions on $F_s$, we have:

$$|F_s(x)| = |F_s(x) - F_s(0) + F_s(0)| \le |F_s(x) - F_s(0)| + |F_s(0)| \tag{36}$$
$$\le \|x - 0\|_\infty + |F_s(0)| \le \|x\|_\infty + \sup_s |F_s(0)|. \tag{37}$$

$\square$

**Fact 2.** *Let $x, q \in \mathbb{R}^d$ be such that $0 \leq q_i \leq c$ for all $i$. Let $\widetilde{x} = \mathbf{clip}_c(x)$. Then, $\|\widetilde{x} - q\|_\infty \leq \|x - q\|_\infty$.*

*Proof.* For any $i \in \{1, \ldots, d\}$, we have $|\widetilde{x}_i - q_i| \leq |x_i - q_i|$, since $0 \leq q_i \leq c$. The result follows. $\qquad\square$

**Proposition 2.** $\forall a, b, c > 0$

$$\lim_{x \to \infty} \frac{1}{x^c} \exp\big(a[\log \log(x^b)]^2\big) = 0$$

*Proof.* We have

$$\frac{1}{x^c} \exp\big(a[\log \log(x^b)]^2\big) = \exp\big(a[\log \log(x^b)]^2 - c \log x\big)$$

$$= \exp\Big(a[\log u]^2 - \frac{c}{b} u\Big), \quad \text{by setting } u = \log(x^b)$$

And, for any $k > 0$, we have

$$\lim_{u \to \infty} \log^2 u - ku = -\infty. \tag{38}$$

which allows us to conclude. $\qquad\square$

**Proposition 3.** *If we set $\delta' = \delta'(\varepsilon) = \varepsilon^5$, we have:*

$$n(\varepsilon, \delta'(\varepsilon)) = \mathcal{O}\Big(\frac{1}{\varepsilon^{4+c}}\Big), \quad \forall c > 0$$

*Proof.* We have:

$$n_{\mathtt{sampleV}}(\varepsilon, \delta'(\varepsilon)) \leq \eta_1 \left[ \log_{\frac{1}{\gamma}}\Big(\frac{\overline{\varepsilon}/\gamma}{\varepsilon}\Big) \right]^{\eta_2(\varepsilon^3)} \frac{1}{\varepsilon^2}$$

$$= \underbrace{\left[ \log_{\frac{1}{\gamma}}\Big(\frac{\overline{\varepsilon}/\gamma}{\varepsilon}\Big) \right]^{\log_2\big(k \log\big(\frac{2K}{\varepsilon^3}\big)\big)}}_{(A)} \frac{1}{\varepsilon^2}$$

where $k$ is a constant that does not depend on $\varepsilon$. The term (A) can be rewritten as:

$$\left[ \log_{\frac{1}{\gamma}}\Big(\frac{\overline{\varepsilon}/\gamma}{\varepsilon}\Big) \right]^{\log_2\big(k \log\big(\frac{2K}{\varepsilon^3}\big)\big)} = \Big[c_1 \log\Big(\frac{c_2}{\varepsilon}\Big)\Big]^{c_3 \log\big[k \log\big(\frac{c_4}{\varepsilon^3}\big)\big]}$$

$$= \exp\Big\{c_3 \log\Big[k \log\Big(\frac{c_4}{\varepsilon^3}\Big)\Big] \log\Big(c_1 \log\Big(\frac{c_2}{\varepsilon}\Big)\Big)\Big\}$$

which can be shown to be $\mathcal{O}\big(\frac{1}{\varepsilon^c}\big)$ for any $c > 0$ by applying proposition 2 after some algebraic manipulations.

Hence,

$$n_{\mathtt{sampleV}}(\varepsilon, \delta'(\varepsilon)) = \frac{1}{\varepsilon^2} \mathcal{O}\Big(\frac{1}{\varepsilon^c}\Big) = \mathcal{O}\Big(\frac{1}{\varepsilon^{2+c}}\Big), \quad \forall c > 0.$$

Since we have

$$n(\varepsilon, \delta') = N(\varepsilon) n_{\mathtt{sampleV}}(\varepsilon, \delta')$$

with $N(\varepsilon) = \widetilde{\mathcal{O}}\big(1/\varepsilon^2\big)$, this proves the result. $\qquad\square$

**Corollary 1.** *If we set $\delta' = \delta'(\varepsilon) = \varepsilon^5$, we have:*

$$\lim_{\varepsilon \to 0} \delta'(\varepsilon) n(\varepsilon, \delta'(\varepsilon)) = 0$$

*Proof.* It is an immediate consequence of proposition 3 by taking $c \in ]0, 1[$.  □

## E   On other smooth approximations of the max

In this paper we focus on the $\mathrm{LogSumExp}_\lambda$ function as a smooth approximation to the maximum function. Yet our proof is more general and can handle any approximation of the max function which verifies the properties listed in Section 5. For instance let's consider the following regularization of the Bellman equation:

$$F(Q) = \max_{(\pi_a)_{a \in \mathcal{A}}} \sum_{a \in \mathcal{A}} (Q_a \cdot \pi_a + \lambda \sqrt{\pi_a}) \tag{39}$$

This smooth function is particularly interesting because it approximates the distribution of pulled armed of the UCB algorithm by taking $\lambda = 2c \cdot \sqrt{\frac{\ln(n)}{n}}$ (see 41 and notice that $\pi_a^\star \cdot n$ approximates $n_a$). We show that this smooth approximation of the maximum verifies the assumptions made in Section 5. We have

$$F(Q) = \sum_{a \in \mathcal{A}} \left( Q_a \cdot \pi_a^\star + \lambda \sqrt{\pi_a^\star} \right) \tag{40}$$

and we can show that $\nabla_Q F(Q) = \pi^\star$. Therefore point 1, 2 and 3 of Section 5 are verified. Now by differentiating with respect to $\pi$ this time:

$$\forall a \in \mathcal{A} \quad Q_a + \frac{\lambda}{2\sqrt{\pi_a^\star}} = U \tag{41}$$

where $U$ is the Lagrange multiplier. Using the fact that $\sum_{a \in \mathcal{A}} \pi_a^\star = 1$, we get

$$\sum_{a \in \mathcal{A}} \left( \frac{\lambda/2}{U - Q_a} \right)^2 = 1 \tag{42}$$

Because $U > \max_a \pi_a^\star$ the derivative of the left side with respect to $U$ is positive for all $Q_a \in [0, (1 + M_\lambda)/(1 - \gamma)]^{|\mathcal{A}|}$. Using the inverse function theorem we get that $U$ is differentiable with respect to Q and that $\pi_a^\star = \left( \frac{\lambda/2}{U - Q_a} \right)^2$ is also differentiable with respect to Q. Finally because $[0, (1 + M_\lambda)/(1 - \gamma)]^{|\mathcal{A}|}$ is compact we can conclude that $F$ is $L$-smooth for some $L \geq 0$ verifying point 4 of Section 5.

## F   Experimental validation of the theoretical results

In this section, we present the experiments we made to verify the correctness of our sample complexity bounds (Theorem 1) and of our consistency results (Theorem 2).

### F.1   Checking the sample complexity guarantee

The key step for proving Theorem 1 is using Lemma 1, that bounds the number of calls to the generative model made by a call to `sampleV`$(s, \varepsilon)$.

Figure 2 shows the simulated number of calls to the generative model made by $n_{\texttt{sampleV}}(\varepsilon, \delta')$ as a function of $1/\varepsilon$ and compares it to our theoretical bound in Lemma 1 and to the number of calls that would be required by a Sparse Sampling strategy, which corresponds to the bound in Proposition 1

extrapolated to all values of $\varepsilon$. The simulated values where obtained by computing the following recurrence for several values of $\varepsilon$:

$$
n_{\mathtt{sampleV}}^{\mathrm{sim}}(\varepsilon, \delta') = \begin{cases} 1 + n_{\mathtt{sampleV}}^{\mathrm{sim}}\left(\frac{\varepsilon}{\sqrt{\gamma}}, \delta'\right) + KN(\sqrt{\kappa}\varepsilon) n_{\mathtt{sampleV}}^{\mathrm{sim}}\left(\sqrt{\frac{\kappa\varepsilon}{\gamma}}, \delta'\right), & \text{if } \varepsilon < \kappa, \\ \gamma^{\frac{1}{2}H(\varepsilon)(H(\varepsilon)-1)}\left(\frac{2\alpha(\delta')}{\varepsilon^2}\right)^{H(\varepsilon)}, & \text{otherwise.} \end{cases}
$$

Figure 3 shows the mumber of calls to the generative model made by $n_{\mathtt{sampleV}}$ as a function of the regularization parameter $\lambda$ in order to achieve a relative error of $0.01$[7] and its ratio with respect to the number of calls that would be required by Sparse Sampling in the same setting. We see that fewer samples are required as the regularization increases. We also see that, for small $\lambda$, there is no advantage with respect to Sparse Sampling, but `SmoothCruiser` has a very large advantage when the regularization $\lambda$ grows.

Figure 2: Simulated number of calls to the generative model made by $n_{\mathtt{sampleV}}(\varepsilon, \delta')$ as a function of $1/\varepsilon$ compared to our theoretical bound (Lemma 1) and to the number of calls that would be required by a Sparse Sampling strategy. The parameters used were: $\gamma = 0.2$, $\delta' = 0.1$, $K = 2$ and $\lambda = 0.1$.

### F.2 Checking the consistency guarantee

Using our MCTS analogy in Section 3.3, the two most computationally costly operations of `SmoothCruiser` are the `selectAction` and the `evaluateLeaf` functions. They both rely on estimates of the $Q$ function with some required accuracy. Hence, for a *sanity-check*, we implemented the function `sampleV` by replacing its calls to `estimateQ`($s$, accuracy) by the true Q function at state $s$ plus some accuracy-dependent noise, and we denote this simplified version of `sampleV` by `sampleV`$_{\mathrm{check}}$ . This allowed us to verify that our bounds for the bias of the `sampleV` outputs (Lemma 2) are correct. After $N_{\mathrm{sim}}$ calls to `sampleV`$_{\mathrm{check}}(\varepsilon, \delta')$, we compute the error

$$
\widehat{\Delta}(s, \varepsilon) = \frac{1}{N_{\mathrm{sim}}} \sum_{i=1}^{N_{\mathrm{sim}}} \left( \widehat{V}_i(s, \varepsilon) - V(s) \right) \tag{43}
$$

Figure 3: Number of calls to the generative model made by $n_{\mathtt{sampleV}}$ as a function of the regularization parameter $\lambda$ in order to achieve a relative error of $0.01$ (left) and its ratio with respect to the number of calls that would be required by Sparse Sampling in the same setting (right). The parameters used were: $\gamma = 0.2$, $\delta' = 0.1$ and $K = 2$.

where $s$ is a reference state and $\widehat{V}_i(s, \varepsilon)$ is the output of the $i$-th call to $\mathtt{sampleV}_{\mathrm{check}}(s, \varepsilon)$. Lemma 2 states that, for some high probability event $B$, we have $-\varepsilon \le \mathbb{E}\left[\widehat{\Delta}(s, \varepsilon) | B\right] \le \varepsilon$. Hence, for large $N_{\mathrm{sim}}$, we should have $-\varepsilon \le \widehat{\Delta}(s, \varepsilon) \le \varepsilon$ approximately.

Table 1 shows simulated values of $\widehat{\Delta}(s, \varepsilon)$ and their standard deviations for different environments. The value of $N_{\mathrm{sim}}$ was chosen so that $\widehat{\Delta}(s, \varepsilon)$ is close to its mean, by using Hoeffdings's inequality and assuming that $\widehat{V}_i(s, \varepsilon)$ is bounded by $C_\gamma$ (which holds with high probability, by Lemma 2).

| Environment | $\widehat{\Delta}(s, \varepsilon)$ |
|---|---|
| 5-Chain | $(-1.21 \pm 1.65) \times 10^{-2}$ |
| 10-Chain | $(-1.20 \pm 1.63) \times 10^{-2}$ |
| 5x5-GridWorld | $(-0.71 \pm 2.04) \times 10^{-2}$ |
| 10x10-GridWorld | $(-0.71 \pm 2.03) \times 10^{-2}$ |

Table 1: Simulated values of $\widehat{\Delta}(s, \varepsilon)$ and its standard deviation for different environments, for $\varepsilon = 0.35$. The value of $\varepsilon$ was chosen such that $\varepsilon \le \kappa/4$ in all environments. The parameters used were: $N_{\mathrm{sim}} = 32723$, $\gamma = 0.2$ and $\lambda = 10$. The $n$-Chain environments have $K = 2$ and $n$ states and the $n \times n$-GridWorld environments have $K = 4$ and $n^2$ states.

The code for the experiments is at https://github.com/omardrwch/smoothcruiser-check.