[Reviews · NeurIPS 2019]

Reviewer 1



An interesting observation made by this paper is that computing entropy-regularized value functions can be seen as a smooth relaxation of a planning problem. The paper is well written and easy to follow.

Reviewer 2



This paper has an interesting algorithmic idea and the properties related to such an algorithm is well investigated. This idea and algorithm should be interesting to the readers. The paper is well written and easy to follow. My main concern of this paper is that it is missing experimental results. As a paper with algorithm invention as its core contribution, I believe experimental results should be crucial to demonstrate the possible future impact and would be important for many readers to understand and follow the work. Any result should be very helpful.

Reviewer 3



This theoretical paper considers the problem of computing optimal value function in entropy-regularized MDPs and two-player games. It shows that the smoothness property of the Bellman operator in the presence of entropy regularized policies (and possibly other forms of regularization), can be used to derive a sample complexity which is polynomial of order O((1/ε)^{4+c}), with c being a problem independent constant and ε the precision of the value function estimate. The proof is built upon the proposed algorithm, SmoothCruiser, an algorithm motivated in the sparse sampling algorithm of Kearns et al that recursively estimates V through samples and subsequently aggregates the results. This sampling dynamic programming is done up to a depth when the required number of samples is no longer polynomial. The paper is very well written and provides a solid result. Despite the practical limitations of the settings, e.g., a model oracle that can sample anywhere in the state-action space, this is a common assumption in theoretical work and the results are significant and deserve publication. It connects the sparse sampling algorithm with the literature on entropy-regularized MDPs. Although this regularization is applied widely in many practical situations and there are a few theoretical results in the direction taken in this work. Besides this model assumption, the little applicability of this result/algorithm is still a concern. Although the constant c is problem independent, the value of \kappa grows very quickly, since it is inversely proportional to the product of K, the number on the of actions, and the strength of the regularization. It would be very nice if the authors can come up with a minimally interesting MDP that can illustrate this in practice. The feeling is that in most of the problems, the interesting region between \lambda infinity and 0 is small, and the algorithm will roll-back to Kearns spase sampling most of the times. I also found a bit stretched the mapping of SmoothCruiser to a generic MCTS algorithm. In MCTS, selecting an action does not involve any sampling. I wonder what is the real motivation of making this parallelism if it is not exploited in the paper. Finally, I would like to hear whether these complexity results can be improved for restricted class of problems, for example, in the case of MDPs or games with deterministic dynamics. ------------- POST-REBUTTAL I read the rebuttal and I am largely satisfied with the authors' replies.

[Author Response · NeurIPS 2019]

We would like to thank the reviewers for their comments and for their positive feedback on our contributions. Although the reviewers expressed their concern regarding the lack of experiments, we would like to stress that it is a theoretical paper, providing an algorithm that has a polynomial sample complexity guarantee for general environments - with a possibly infinite state space and stochastic transitions - *for which there was no known polynomial bound*. It also highlights a theoretical benefit of regularization in MDPs, which has been empirically studied in several recent works on reinforcement learning. The work of Kearns et al. (1999), which is very related to our setting, is also purely theoretical. Their work later inspired other algorithms that can be used in practice, such as UCT (Kocsis and Szepesvári, 2006).

Nonetheless, to see the tightness of our guarantees, we had done the experiments described below. We omitted them from the submission but we can include them in the appendix if the reviewers find them useful:

- Using our MCTS analogy in Section 3.3, the two most computationally costly operations of SmoothCruiser are the `selectAction` and the `evaluateLeaf` functions. They both rely on estimates of the Q function with some required accuracy. Hence, for a *sanity-check*, we replaced `estimateQ`($s$, accuracy) by the true Q function at state $s$ plus some accuracy-dependent noise. This allowed us to verify that our bounds for the bias of the `sampleV` outputs are correct (Lemma 2). For instance, in a 3x3 GridWorld with $\gamma = 0.5$ and $\lambda = 0.1$, averaging $10^3$ calls to sampleV gives an error of about $(0.98 \pm 4.53) \times 10^{-4}$, while our bounds predict a mean error of at most $7 \times 10^{-4}$.
- Using small model parameters ($\gamma = 0.2$, $\delta = 0.9$, $K = 2$, $\lambda = 0.01$), we checked that our sample complexity bound is tight, by simulating the recurrence relation that led to our bound. This is illustrated on the leftmost figure below.

We now respond to specific points raised by the reviewers:

- **[R3] I also found a bit stretched the mapping of SmoothCruiser to a generic MCTS algorithm. In MCTS, selecting an action does not involve any sampling.** We make connection with MCTS because SmoothCruiser can be seen as form of MCTS with sampling (such as AlphaGo, which is also a sampling-based MCTS algorithm). In addition, the analogy to MCTS could be useful to inspire more practical versions of SmoothCruiser—this is the purpose of Section 3.3.
- **[R2 and R3] [R3] In most of the problems, [...] the algorithm will roll-back to Kearns et al. (1999) sparse sampling most of the times. [R2] Comparison to other algorithms.** Indeed, the interesting regime of SmoothCruiser is when the required precision $\varepsilon$ is smaller than $\kappa$, which is proportional to the regularization strength ( $\lambda = 1/L$) and inversely proportional to the number of actions. Hence, in weakly regularized problems with a lot of actions, SmoothCruiser will only be beneficial if $\varepsilon$ is very small (as we can see in the leftmost figure below, the sample complexity for $1/\varepsilon > 1/\kappa$ increases much more *slowly* than when $1/\varepsilon \leq 1/\kappa$). However, we can choose the regularization constant to make $\kappa$ big, which would reduce the sample complexity of the algorithm. This tradeoff between sample complexity and regularization is another interesting observation in our work. We propose to include in the paper the middle and the rightmost figures below that illustrate this idea. For each $\lambda$ (x axis) we plot in the middle figure how many samples SmoothCruiser requires to achieve a relative error of $0.01$ (normalized by the value function upper bound). We see that fewer samples are required as the regularization increases. The rightmost figure shows the same thing, but normalized by the number of samples required by the sparse sampling strategy: we see that, for small $\lambda$, there is no advantage with respect to sparse sampling, but SmoothCruiser has a very large advantage when $\lambda$ grows.
- **[R3] I found the title a bit misleading [...]a weak connection with MCTS, I would not use planning in the title.** The word 'planning' in the title agrees with a what we think is a commonly used definition: a procedure that takes a model as input and returns the value function as output [e.g., Sutton and Barto, 2nd edition, Section 8.1]. 'Planning' is also used in the title of Kearns at al. (1999), which has the same setting as us. We welcome a suggestion though.
- **[R3] I would like to hear whether these complexity results can be improved for restricted class of problems, for example, in the case of MDPs or games with deterministic dynamics.** Our algorithm focuses on very general environments for which there were *no previously known polynomial bounds*. For some restricted classes of problems, algorithms with polynomial guarantees already exist, for instance, by Hren and Munos, 2008 (deterministic environments) and Bubeck and Munos, 2010 (deterministic transitions). In our case, deterministic rewards and transitions could only reduce the value of $c_1$ in Theorem 1, by setting $N(\varepsilon) = 1$ in algorithm 2 if $\varepsilon \geq \kappa$.



[Meta-Review · NeurIPS 2019]

The reviewers were in consensus that this is an interesting and well written paper with a significant theoretical contribution. While empirical results should not be strictly required for a paper that is strong theoretically, they would nonetheless greatly improve the paper, and thus the authors are strongly encouraged to include them in the final version, even if they are relegated to supplementary material.